# Towards Single-Source Domain Generalized Object Detection via Causal Visual Prompts

**Chen Li** [1][†]   **Huiying Xu** [2][†]   **Changxin Gao**[1][*]   **Zeyu Wang** [3]   **Yun Liu** [4]   **Xinzhong Zhu** [2]

[1] National Key Laboratory of Multispectral Information Intelligent Processing Technology,
School of Artificial Intelligence and Automation, Huazhong University of Science and Technology,
[2] Zhejiang Key Laboratory of Intelligent Education Technology and Application,
Zhejiang Normal University, [3] Northwest Polytechnical University, [4] Nankai University.
{lichenrui27, cgao}@hust.edu.cn

## Abstract

Single-source Domain Generalized Object Detection (SDGOD), as a cutting-edge research topic in computer vision, aims to enhance model generalization capability in unseen target domains through single-source domain training. Current mainstream approaches attempt to mitigate domain discrepancies via data augmentation techniques. However, due to domain shift and limited domain-specific knowledge, models tend to fall into the pitfall of spurious correlations. This manifests as the model's over-reliance on simplistic classification features (e.g., color) rather than essential domain-invariant representations like object contours. To address this critical challenge, we propose the Cauvis (**Cau**sal **Vis**ual Prompts) method. First, we introduce a Cross-Attention Prompts module that mitigates bias from spurious features by integrating visual prompts with cross-attention. To address the inadequate domain knowledge coverage and spurious feature entanglement in visual prompts for single-domain generalization, we propose a dual-branch adapter that disentangles causal-spurious features while achieving domain adaptation via high-frequency feature extraction. Cauvis achieves state-of-the-art performance with 15.9–31.4% gains over existing domain generalization methods on SDGOD datasets, while exhibiting significant robustness advantages in complex interference environments.

## 1 Introduction

Autonomous driving object detection models face challenges from complex targets and variable weather conditions, where domain shifts significantly degrade detection accuracy. Compared to open-set [1] or few-shot object detection [2], Single-domain Generalized Object Detection (SDGOD) [3] — which relies solely on single-source domain data for training without cross-domain prior knowledge (neither multi-domain training data nor target domain prompts) — exhibits severe overfitting when encountering domain shifts. This has become a critical bottleneck restricting the practical deployment of autonomous driving perception systems.

The core challenge of SDGOD lies in achieving precise multi-object localization and recognition under multi-domain scenarios. Machine learning models in such tasks are prone to training data bias, over-relying on spurious correlations between target labels and background noise/secondary features. These spurious correlations arise through two interconnected mechanisms: spatially, as training data predominantly position vehicles in central image regions (characteristic of main lane scenarios in Cityscapes [4]) and pedestrians near edges (typical sidewalk areas), models erroneously establish

---

[†]Equal contributions.
[*]Corresponding Author. Project Link: https://github.com/lichen1015/Cauvis

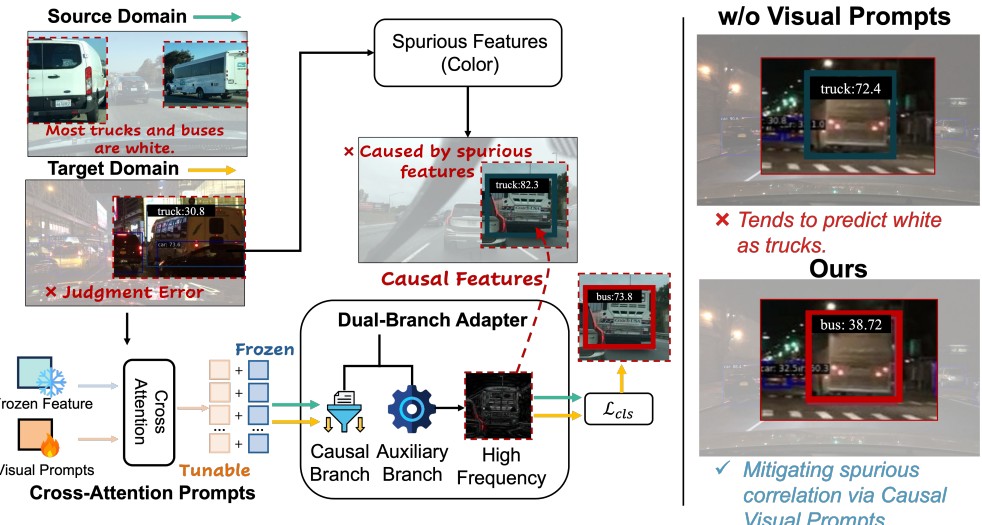

Figure 1: The interference of spurious features (such as color) between the source domain and the target domain causes existing methods [7] to be prone to errors in the target domain, often mistakenly identifying the color white as trucks, showing poor generalization ability. Our method, leveraging Causal Visual Prompts combined with a branch that can extract high-frequency features, effectively stops the model from making predictions based on spurious features, improving its accuracy in the target domain. "Ice" signifies frozen parameters, while "fire" indicates fine-tunable parameters.

"vehicle-center" spatial bindings while rigidly anchoring pedestrians to peripheral zones, thus ignoring natural variations in real-world object distributions. In parallel, color biases emerge through misguided color-category associations, specifically linking white hues to buses/trucks (common in BDD100K urban transit liveries [5]) and black objects to cars. Crucially, such spurious correlations based on superficial features lead to significant degradation in the model's generalization performance in Out-of-Distribution (OOD) scenarios. Although recent work like UFR [6] attempts to address scene confounders through causal attention learning, their reliance on Faster R-CNN [7] frameworks and heuristic perturbation strategies limits the theoretical grounding in disentangling causal features.

While existing studies leverage Domain-Invariant Representation theory [3] to enhance OOD generalization, the systematic analysis of spurious correlations remains insufficiently addressed. Current methodologies [3, 6, 8, 9] primarily pursue two complementary strategies: domain-invariant constraints that architecturally disentangle invariant features from domain-specific characteristics [6], and data augmentation approaches [3, 9], which simulate diverse domain distributions to suppress biased feature dependencies through extended feature space coverage. Nevertheless, these empirically-driven solutions lack unified theoretical grounding, often attributing model deficiencies to symptomatic biases (data skews/attention misallocations/prototype distortions) rather than addressing the fundamental issue – **the inherent prevalence of spurious correlations within single-domain training data**.

To address these issues, we propose Cauvis (Causal Visual Prompts) method whose core innovations include: Cross-Attention Prompts that mathematically establish the equivalence between visual prompts and backdoor adjustment operations [10] in causal inference, providing theoretical foundations for spurious correlation mitigation; and designing a Dual-Branch Adapter that disentangles causal/spurious features via systematic representation decoupling, while enhancing cross-domain knowledge coverage. Specifically, it integrates Fourier transform to achieve causal feature decoupling through high-frequency component extraction, while incorporating nonlinear feature modeling modules to enhance the prompt's domain distribution coverage capabilities.

Our principal contributions are:

- We are the first to introduce DINOv2 [11] as the backbone in the SDGOD. By freezing parameters to reduce training costs, extensive experiments validate that this approach significantly enhances detector performance.

- Through theoretical analysis and experimental observations, we connect spurious correlations to the decline in generalization performance, establishing a new theoretical framework for SDGOD.
- We propose Cross Attention Prompts with theoretical analysis demonstrating their equivalence to backdoor adjustment mechanisms for suppressing spurious correlations. Furthermore, we design a Dual-branch Adapter that addresses the limitations of conventional methods in domain knowledge coverage and feature decoupling through explicit causal relationship modeling.

## 2    Related Work

**Single Domain Generalized Object Detection.** In the field of SGDOD, existing approaches primarily focus on two strategies: 1) imposing domain-specific constraints to extract domain-invariant representations, and 2) enhancing input diversity through systematic data augmentation. CDSD [3] introduces a cyclic-disentangled self-distillation framework to decouple domain-invariant representations through multi-stage loss constraints. OA-DG [12] employs multi-level transformations and object-aware mixing strategies to reduce inter-domain representational discrepancies. SRCD [9] constructs self-augmented compound domains to mitigate category-background bias. UFR [6] leverages physics-inspired data augmentation to simulate potential domain distributions. ClipGap [13] utilizes cross-modal prompting from the CLIP model [14] to achieve semantics-guided enhancement. G-NAS [8] optimizes domain-invariant constraints via differentiable neural architecture search. Our work achieves efficient generalization without introducing artificially synthesized training samples.

**Visual Prompts.** To mitigate cross-domain discrepancies, visual prompting (VP) [15], inspired by prompt learning in NLP, has emerged as an efficient adaptation paradigm. It introduces lightweight tunable parameters, typically optimizing only 1% of model weights, to align with downstream tasks. In few-shot tasks, VP [16] enhances model adaptability to novel categories via sparsely annotated perturbation mechanisms like spatial attention-guided local enhancement. Visual Prompt Tuning (VPT) [17], a representative approach, embeds learnable prompt vectors across Transformer layers to enable hierarchical feature interaction for parameter adaptation. Rein [18], an efficient parameter tuning method, is essentially a unique form of visual prompt. These conventional VP methods [17–19] couple prompt features with frozen backbones without explicit causal/spurious disentanglement mechanisms.

Our approach fundamentally diverges from NLP prompting mechanisms: While NLP utilizes discrete semantic operations (e.g., "MASK" token prediction) for textual reconstruction, visual prompts operate in continuous pixel space through frequency-domain perturbations or adversarial noise injection to activate cross-domain knowledge in pretrained VFMs.

## 3    Motivation

### 3.1    Experimental Demonstration of Spurious Correlations

In image classification tasks [20–22], models often suffer from degraded OOD generalization due to over-reliance on spurious correlations in training data. This issue is particularly critical in dense detection tasks: when models focus on non-causal domain-specific features (e.g., background or color) that coincidentally correlate with labels, rather than intrinsic causal features of objects, their performance significantly deteriorates in unseen domains.

To clearly illustrate this issue, we select the bus and truck categories as representative samples for analysis. These categories often have the same color across domains, appearing predominantly white. (see Fig. 1). To create a controlled experimental environment, we crop images of these categories to task-related areas and filter out images with numerous other detection objects, simplifying experimental conditions.

In the experimental design, we artificially increase the probability $p_i$ of the color-category association (i.e, raising the probability of white color in the truck category to $p_i$) and ensure the non-white distribution of the other category (bus) is $1 - p_i$. Based on varying bias strengths, we create several datasets with different bias levels ($p_i \in [0.75, 0.8, 0.85, 0.9]$) and use the original test data as an unbiased test. We train and test on four biased datasets and then retest the results on the unbiased

Table 1: Comparison of DINO [23] and with prompts on SDGOD [3] and CD-FSOD [2].

| Backbone | Method | Zero-Shot (SDGOD) | | | | | Few-Shot (CD-FSOD) | | |
|---|---|---|---|---|---|---|---|---|---|
| | | DC | DF | DR | NR | NC | 1-Shot | 5-Shot | 10-Shot |
| DINOv2 [11] (Large) | DINO[23] | 66.9 | 50.5 | 57.0 | 42.1 | 54.8 | 16.3 | 32.6 | 34.3 |
| | Prompts | 67.4↑+0.5 | 51.7↑+1.2 | 57.3↑+0.3 | 41.0↓-1.1 | 57.4↑+2.6 | 20.4↑+4.1 | 33.9↑+1.3 | 37.1↑+2.7 |

Figure 2: Left: Visualization results of DINOv2 [11] and Cauvis, along with their heatmaps and the differences between. Right: (a) performance of Bus and Truck on the biased and unbiased test datasets. (b) Cauvis exhibits stronger activations both within the object's interior and outside region.

dataset. As shown in Figure 2, the model's dependence on color features evolves. When $p_i$ rises from 0.75 to 0.9, experimental accuracy for trucks rises by 4.2%, but the Truck mAP on the unbiased test set drops by 21%, indicating stronger model reliance on color shortcuts.

Notably, the model achieves higher mAP on biased test sets as $p_i$ increases, suggesting its growing reliance on simplistic **color-category correlations** for decision-making. Our control experiments show that detection models may overly depend on color or background correlations, especially when training data has systematic bias. While a comprehensive quantification of this issue is challenging, these observations offer crucial insights and reveal clues previously overlooked in research.

Visual prompts provide **semantic constraints** beyond pixels. By describing objects' essential attributes, like their geometry or structured volume, they guide models to form domain-invariant causal representations. This helps models avoid relying on statistical shortcuts in training data.

## 3.2 Visual Prompts and Causal Representation

From the perspective of causal representation learning, we can decompose input feature representations into causal $f_C(x)$ and spurious parts $f_S(x)$. The goal is to make the model sensitive to $f_C(x)$ but robust to or capable of ignoring $f_S(x)$. A common strategy is to introduce **interventions** by applying perturbations $\delta \sim p(\Delta)$ to the inputs, simulating observations in different environments—for instance, by artificially altering the background or adding noise ($x' = x + \delta$), thus forcing the model to focus on stable features $f_C(x + \delta)$ [24, 25].

Existing work [24] shows that if a model can only use purely causal information (e.g., representations of foreground regions) for prediction, this is equivalent to performing a causal intervention on the system, which can greatly boost performance under out-of-distribution (OOD) scenarios. For example, the CFA [24] does this by training only on image representations containing foreground (causal features), equivalent to performing a causal intervention. This effectively reduces the influence of the spurious features on predictions.

Visual prompts [26] are directly introduced into the model as additional input parameters via random initialization (e.g., by concatenation). At initialization, these prompt vectors $\delta$ are just random noise, but during training, they are updated by back-propagation to minimize a causal-invariance loss on $f_C(x)$. In effect, the optimization pushes $\delta$ to lie in the same low-dimensional causal subspace spanned by $f_C$, and away from spurious directions. Any prompts that fail to drive $f_C(x + \delta) \approx f_C(x)$ become suboptimal and can be pruned or re-initialized. Under the causal invariance assumption (see

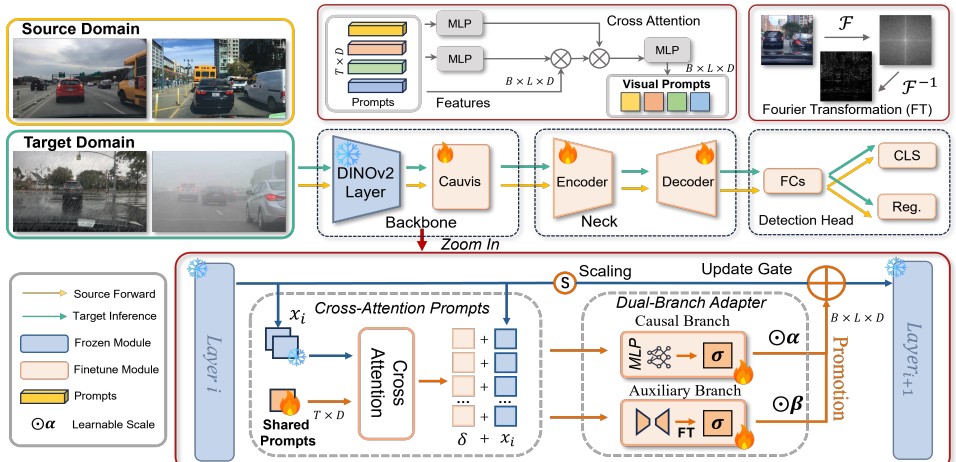

Figure 3: Overview of the Cauvis. The top part illustrates the Cross Attention mechanism and the Fourier Transformation. The bottom part shows the detailed structure of Cauvis, including Cross-Attention Prompts and Dual-Branch Adapter. The Cauvis module is integrated into each layer of the backbone. Blue for frozen parameters, orange for fine-tuning parameters.

the Appendix C), when the prompts converge to their optimal value $p^*$, we have

$$\left.\frac{\partial f_C(x+\delta)}{\partial \delta}\right|_{p=p^*} = 0. \tag{1}$$

i.e., the gradient of the causal feature under this perturbation vanishes, indicating that the prompts do not disrupt the original causal structure [27]. Further, by applying a first-order Taylor expansion, we obtain $f_C(x+\delta) = f_C(x) + o(\delta)$, which indicates that when the perturbation $\delta$ is sufficiently small, the change in the causal feature $f_C$ is negligible. In other words, the tiny disturbance introduced by the prompt does not disrupt the original causal structure.

From the frequency-domain perspective, $\mathcal{F}$ and $\mathcal{F}^{-1}$ is Fourier and its inverse. we can express:

$$f_C(x) = \mathcal{F}^{-1}\big(H_{\text{causal}} \odot \mathcal{F}(x)\big). \tag{2}$$

The perturbations introduced by the prompt are primarily concentrated in the **low-frequency region**, and the filter function $H_{\text{causal}}(\omega)$ has a response that approaches zero for these low-frequency components. Therefore, it likewise does not affect the resulting causal features. We model visual prompts (initialized to zero) as random perturbations applied to the image's low-frequency components. Through an analysis of causal invariance, we find that they **suppress spurious features while preserving causal information**. Moreover, by further incorporating causal inference techniques such as back-door adjustment [10], they can strengthen those causal features.

# 4 Methods

We present Cauvis, which includes Cross-Attention Prompts (Section 4.1) and Dual-Branch Adapter (Section 4.2), as shown in Fig. 3. It uses learnable parameters to dynamically adjust the fusion ratio of pre-trained knowledge and visual prompts.

## 4.1 Cross-Attention Prompts

We propose Cross-Attention Prompts to address limitations of existing methods in causal reasoning and domain generalization. Our framework (illustrated in Fig. 1) uses cross-attention to link prompts with features. Unlike prior work (e.g., Rein [18]'s token-based prompts, EVP [28]'s peripheral embeddings, SPT [19]/VPT [17] that relies on dataset-driven prompt initialization), we treat prompts as pseudo-modal signals independent of the image domain. This design thus enables causal intervention via cross-attention.

According to Pearl's causal theory [10], intervening on $X$ yields an outcome distribution $P(\hat{y}|\text{do}(X))$. When a confounder $z$ blocks all non-causal $X \to \hat{y}$ paths, the back-door adjustment formula:

$$P(\hat{y}|\text{do}(X)) = \sum_z P(\hat{y}|X,z)P(z). \tag{3}$$

This adjusts for confounders $z$ to eliminate non-causal dependencies.

In each attention layer, we consider the attention weight matrix $A \in \mathbb{R}^{T \times T}$ (batch dimension omitted) and compute its singular value decomposition (SVD), $A = U\Sigma V^\top$, where $U, V \in \mathbb{R}^{T \times T}$ and $\Sigma$ is diagonal with nonnegative singular values. We perform Singular Value Decomposition (SVD) on $A = U\Sigma V^\top$, where $U \in \mathbb{R}^{T \times T}$ and $V \in \mathbb{R}^{T \times T}$ are orthogonal matrices ($T$ is the token length); $\Sigma = \text{diag}(\sigma_1, \ldots, \sigma_T)$, and $\sigma_1 \geq \sigma_2 \geq \cdots \geq 0$. The SVD provides an orthogonal basis of "directions" in the attention map, where each singular value $\sigma_i$ indicates the importance or strength of that component, in data science terminology, the largest singular values capture dominant representations in $A$. Under our assumption that the causal features in the model's representations form a low-dimensional subspace, we expect them to induce large-magnitude components in $A$. In contrast, spurious or noisy correlations should only contribute smaller singular values. This intuition aligns with recent analysis of Transformer attention: Franco et al. [29] hypothesized that when attention heads focus on specific low-dimensional features, e.g., an "indirect object identification" task, only a few singular values become large, resulting in a **sparse decomposition**.

Based on this, we explicitly partition $\Sigma$ into **causal** and **non-causal** subspaces. Specifically, let $\Sigma_c = \text{diag}(\sigma_1, \ldots, \sigma_k)$ contain the $k$ largest singular values, and $\Sigma_{c\perp} = \text{diag}(\sigma_{k+1}, \ldots, \sigma_T)$ contain the remaining ones. $T$ denotes the dimension (or the maximal rank) of the matrix (or feature space). We then write:

$$A_c = U \begin{pmatrix} \Sigma_c & 0 \\ 0 & 0 \end{pmatrix} V^\top, \qquad A_{c\perp} = U \begin{pmatrix} 0 & 0 \\ 0 & \Sigma_{c\perp} \end{pmatrix} V^\top, \tag{4}$$

$$A = A_c + A_{c\perp}, \qquad A = U_c \Sigma_c V_c^\top + U_{c\perp} \Sigma_{c\perp} V_{c\perp}^\top. \tag{5}$$

Intuitively, $A_c$ projects attention onto the low-rank subspace spanned by the top-$k$ singular vectors (the **causal subspace**), while $A_{c\perp}$ contains residual "noise" components. Crucially, instead of fixing $k$ via hard thresholding, we integrate this decomposition into training: the optimization process implicitly determines the effective rank. By penalizing the magnitude of $\Sigma_{c\perp}$, we encourage the model to place most of its weight in $A_c$, avoiding manual truncation. This objective acts as a **soft thresholding** operator on singular values.

Overall, causal features correspond to **maximum singular value directions** in the column space of $A$, while prompt tuning enhances robustness by suppressing perturbation-sensitive directions associated with smaller singular values (i.e., suppressing spurious features). Through visual prompts, the causal features are strengthened, causing the SVD of matrix $A$ to gradually satisfy $\Sigma_{c\perp} \to 0$, leaving only $\sigma_1, \ldots, \sigma_k$. At this time $V = V_c$, the attention update simplifies to :

$$AV = (U_c \Sigma_c V_c^\top)V = U_c \Sigma_c = \sum_{i=1}^{k} \sigma_i u_i, \tag{6}$$

**Ideal target and conditions.** In probabilistic terms, our optimization *aims to satisfy* the following ideal identity:

$$\mathbb{E}_{z \sim P(z)}\big[f(X, z)\big] = \sum_{i=1}^{k} u_i \, \sigma_i. \tag{7}$$

This equality holds under the theoretical condition that the learned representation is isomorphic to the causal subspace, i.e., the model perfectly aligns $u_i$ with the top-$k$ singular directions that capture causal features. In practice, we optimize *toward* this equality. Here, $\mathcal{Z}$ denotes the confounder space and $z$ an individual confounder. Under this notation, the back-door adjustment in Eq. (3) can be rewritten as:

$$P(\hat{y} \mid do(X)) = \mathbb{E}_{z \sim P(z)}\big[P(\hat{y} \mid X, z)\big] = \mathbb{E}_{z \sim P(z)}\big[f(X, z)\big]. \tag{8}$$

An isomorphism holds when $f(X, z_i) = u_i$ (see Eq. (7) for the definition of $u_i$). When the learned prompt directions span the confounder space $\mathcal{Z}$, *cross-attention between the prompts and the image*

Table 2: Quantitative results (%) on target domain datasets. Bolded represents the best possible outcome, and the underline represents the second-best result.

| Method / mAP | Day Foggy | | | | Dusk Rainy | | | | Night Rainy | | | | Night Clear | | | |
|---|---|---|---|---|---|---|---|---|---|---|---|---|---|---|---|---|
| | $hev._m$ | $mid_m$ | $non_m$ | $all_m$ | $hev._m$ | $mid_m$ | $non_m$ | $all_m$ | $hev._m$ | $mid_m$ | $non_m$ | $all_m$ | $hev._m$ | $mid_m$ | $non_m$ | $all_m$ |
| IterNorm [30] | 25.7 | 33.4 | 27.0 | 28.5 | 34.3 | 25.0 | 13.7 | 22.8 | 19.5 | 11.5 | 8.8 | 12.6 | 38.3 | 27.4 | 25.3 | 29.6 |
| SW [31] | 27.1 | 34.9 | 33.8 | 32.2 | 37.0 | 30.3 | 16.6 | 26.3 | 20.0 | 13.9 | 9.4 | 13.7 | 39.5 | 33.2 | 29.6 | 33.4 |
| FR [7] | 30.6 | 37.8 | 32.6 | 33.5 | 34.6 | 32.0 | 21.1 | 28.0 | 18.7 | 17.5 | 9.0 | 14.2 | 43.3 | 33.7 | 32.2 | 35.8 |
| CDSD [3] | 28.5 | 39.3 | 32.9 | 33.5 | 38.9 | 32.2 | 18.5 | 28.2 | 21.8 | 19.7 | 11.2 | 16.6 | 42.0 | 35.2 | 34.0 | 36.6 |
| DINO [23] | 26.4 | 39.9 | 37.9 | 35.2 | 37.1 | 36.5 | 20.5 | 29.8 | 18.7 | 17.1 | 9.0 | 14.1 | 42.5 | 41.6 | 31.2 | 37.4 |
| SRCD [9] | 32.1 | 41.9 | 34.5 | 35.9 | 40.0 | 31.3 | 19.7 | 28.8 | 25.3 | 16.6 | 11.9 | 17.0 | 43.0 | 36.2 | 32.9 | 36.7 |
| ClipGap [13] | 30.7 | 46.0 | 38.9 | 38.6 | 40.1 | 38.8 | 22.8 | 32.3 | 25.8 | 22.7 | 11.3 | 18.7 | 40.3 | 38.6 | 33.5 | 36.9 |
| G-NAS [8] | 28.5 | 44.8 | 36.1 | 36.4 | 45.2 | 40.6 | 24.7 | 35.1 | 25.0 | 19.3 | 11.1 | 17.4 | 47.4 | 47.0 | 42.2 | 45.0 |
| PDDOC [32] | 31.2 | 45.9 | 39.7 | 39.0 | 41.7 | 40.7 | 23.9 | 33.7 | 24.3 | 23.0 | 13.1 | 19.1 | 41.9 | 40.2 | 35.1 | 38.5 |
| UFR [6] | 32.2 | 47.7 | 39.2 | 39.6 | 40.5 | 42.2 | 22.4 | 33.2 | 26.6 | 22.8 | 11.8 | 19.2 | 45.6 | 40.4 | 37.9 | 40.8 |
| FR [7]+DINOv2 [11] | **51.0** | 56.0 | 46.8 | 50.6 | 63.9 | 62.6 | 53.4 | 59.0 | 55.9 | 44.2 | 34.6 | 43.4 | 59.4 | 56.2 | 52.3 | 55.4 |
| w/o Cauvis | 47.5 | 56.2 | 48.7 | 50.5 | 65.2 | 62.2 | 48.2 | 57.0 | 57.8 | 38.5 | 34.1 | 42.2 | 60.3 | 54.3 | 51.5 | 54.8 |
| FR [7]+Cauvis | 50.5 | 58.0 | 49.6 | 52.3 | 67.0 | 66.8 | 53.9 | 61.3 | 55.2 | 46.1 | 35.3 | 44.1 | 60.9 | 58.7 | 55.7 | 58.3 |
| **Cauvis(Ours)** | 49.1 | **62.2** | **57.7** | **56.5** | **68.7** | **68.8** | **59.2** | **64.6** | **59.0** | **46.8** | **40.6** | **47.6** | **63.5** | **59.4** | **61.0** | **61.2** |

Table 3: Quantitative results (%) on Cityscapes-C (level-5). mPC is an average performance of 15 corruption types. For a fair comparison, Cauvis used FasterRCNN [7] as the base detector.

| Methods | Norm. | Cityscapes→Noise | | | Cityscapes→Blur | | | | Cityscapes→Weather | | | | Cityscapes→Digital | | | | mPC |
|---|---|---|---|---|---|---|---|---|---|---|---|---|---|---|---|---|---|
| | | Guass | Shot | Impul | Defoc | Glass | Motion | Zoom | Snow | Frost | Fog | Bright | Contr | Elas | Pixel | JPEG | |
| FasterRCNN [7] | 42.2 | 0.5 | 1.1 | 1.1 | 17.2 | 16.5 | 18.3 | 2.1 | 2.2 | 12.3 | 29.8 | 32.0 | 24.1 | 40.1 | 18.7 | 15.1 | 15.4 |
| AutoAug [33] | 42.4 | 0.9 | 1.6 | 0.9 | 16.8 | 14.4 | 18.9 | 2.0 | 1.9 | 16.0 | 32.9 | 35.2 | 26.3 | 39.4 | 17.9 | 11.6 | 15.8 |
| AugMix [34] | 39.5 | 5.0 | 6.8 | 5.1 | 18.3 | 18.1 | 19.3 | 6.2 | 5.0 | 20.5 | 31.2 | 33.7 | 25.6 | 37.4 | 20.3 | 19.6 | 18.1 |
| Stylized [35] | 36.3 | 4.8 | 6.8 | 4.3 | 19.5 | 18.7 | 18.5 | 2.7 | 3.5 | 17.0 | 30.5 | 31.9 | 22.7 | 33.9 | 22.6 | 20.8 | 17.2 |
| OA-Mix [12] | 42.7 | 7.2 | 9.6 | 7.7 | 22.8 | 18.8 | 21.9 | 5.4 | 5.2 | 23.6 | 37.3 | 38.7 | 31.9 | 40.2 | 22.2 | 20.2 | 20.8 |
| SupCon [36] | 43.2 | 7.0 | 9.5 | 7.4 | 22.6 | 20.2 | 22.3 | 4.3 | 5.3 | 23.0 | 37.3 | 38.9 | 31.6 | 40.1 | 24.0 | 20.1 | 20.9 |
| FSCE [37] | 43.1 | 7.4 | 10.2 | 8.2 | 23.3 | 20.3 | 21.5 | 4.8 | 5.6 | 23.6 | 37.1 | 38.0 | 31.9 | 40.0 | 23.2 | 20.4 | 21.0 |
| OA-DG [12] | 43.4 | 8.2 | 10.6 | 8.4 | 24.6 | 20.5 | 22.3 | 4.8 | 6.1 | 25.0 | 38.4 | 39.7 | 32.8 | 40.2 | 23.8 | 22.0 | 21.8 |
| FR [7]+DINOv2 [11] | 44.0 | 14.5 | 16.3 | 13.5 | 33.9 | 27.3 | 33.1 | 13.7 | 25.2 | 31.5 | 39.5 | 42.5 | 38.9 | 42.1 | 35.9 | 31.8 | 29.3 |
| **Cauvis(Ours)** | **54.6** | **16.8** | **19.8** | **15.2** | **41.4** | **34.0** | **39.2** | **15.8** | **29.8** | **36.7** | **48.8** | **53.0** | **49.5** | **52.0** | **43.9** | **38.8** | **35.6** |

Table 4: Quantitative results (%) on BDD100K-C (level-5). mPC is an average performance of 15 corruption types.

| Methods | Norm. | Cityscapes→Noise | | | Cityscapes→Blur | | | | Cityscapes→Weather | | | | Cityscapes→Digital | | | | mPC |
|---|---|---|---|---|---|---|---|---|---|---|---|---|---|---|---|---|---|
| | | Guass | Shot | Impul | Defoc | Glass | Motion | Zoom | Snow | Frost | Fog | Bright | Contr | Elas | Pixel | JPEG | |
| FR [7] + DINOv2 [11] | 53.2 | 33.4 | 35.1 | 32.3 | 42.5 | 38.5 | 41.3 | 20.7 | 39.4 | 38.3 | 51.3 | 51.0 | 51.1 | 49.1 | 48.4 | 47.3 | 41.3 |
| FR [7] + Cauvis | **55.3** | **34.4** | **36.5** | **33.3** | **44.3** | **41.4** | **43.1** | **21.6** | **41.1** | **42.3** | **54.8** | **53.9** | **54.0** | **51.2** | **51.3** | **50.4** | **43.6** |

*features* explicitly implements the $do(z)$ operator: the prompts supply the directions, while the cross-attention weights gate feature components, suppressing those aligned with non-causal $z$-variations and driving the corresponding singular values toward zero ($\sigma_{k+1}, \ldots, \sigma_T \to 0$).

Under frozen-backbone visual prompt learning, the cross-attention achieves **statistical equivalence** to causal intervention via two key mechanisms: Dominant singular vectors $\{u_1, ..., u_k\}$ span a low-dimensional subspace capturing causal features, while smaller $\sigma_i$ (associated with $z_{k+1}, ..., z_T$) encode confounding noise. By optimizing prompts to maximize $\sum_{i=1}^{k} \sigma_i^2$, cross-attention suppresses confounding directions $\{u_{k+1}, ..., u_T\}$, effectively performing $do(z_i = 0)$ for $i > k$. This provides a unified theoretical framework for interpretable domain generalization: cross-attention act as **back-door adjustment** that disentangle $\mathcal{Z}$-induced spurious correlations.

## 4.2 Dual-Branch Adapter

One hand, the prompts need to include all confounders $z_i$. On the other hand, compared to few-shot cross-domain object detection (CD-FSOD [2]), SDGOD methods face a steeper challenge due to the complete absence of target domain data, which severely limits the performance boost from visual prompts. As shown in Table 1, SDGOD only achieves a +2.6% improvement in the NC, versus +4.1% under 1-shot. This highlights the need for extra adapters to close the generalization gap in zero-shot tasks. To address this, we propose a dual-branch adapter (see Fig. 3).

**Causal Branch** focuses on local spatial patterns and causal semantics. It uses an MLP to map the *cross-attended visual prompt activations* produced by the Cross-Attention Prompts (CAP) to causal features. Let $\tilde{p}_i = \text{CAP}(p_i, X)$ denote the activation obtained from the raw prompt parameter $p_i$ and image features $X$. The mapping is formalized as

$$y_i = \sigma\big(\text{MLP}(\tilde{p}_i)\big), \tag{9}$$

where $\sigma$ is a sigmoid applied elementwise, so that $y_i \in [0, 1]^d$.

**Auxiliary Branch** leverages Fourier analysis. It extracts domain-invariant features by focusing on phase information or filtering the amplitude. This approach, as Yang et al. [38] suggest, isolates stable structures from domain-specific artifacts. We apply Fourier transforms to suppress domain-dependent

Table 5: Detailed ablation study in Cauvis.

| Configuration | | SDGOD (mAP) | | | | | |
|---|---|---|---|---|---|---|---|
| Component | Variant | DC | DF | DR | NR | NC | Avg. |
| Full Model (**Cauvis**) | | **73.7** | **56.5** | **64.6** | 47.6 | **61.2** | **60.7** |
| w/o Dual Branch Adapter | | 73.0 | 55.1 | 62.8 | **48.5** | 59.4 | 59.8 |
| w/o Cross-Attention Prompts | | 70.5 | 53.3 | 60.0 | 45.5 | 57.6 | 57.4 |
| w/o Visual Prompts | | 66.9 | 50.5 | 57.0 | 42.1 | 54.8 | 54.3 |
| w/o DINOv2 [11] (Baseline) | | 54.5 | 35.2 | 29.8 | 14.1 | 37.4 | 34.2 |
| Causal Branch | ▷ Multi-head | 72.8 | 55.2 | 62.2 | 46.4 | 60.6 | 59.4 |
| | ▷ SE Block | 71.4 | 54.9 | 62.4 | 47.0 | 58.9 | 58.9 |
| | ▷ Gate Unit | 72.9 | 55.3 | 62.1 | 46.0 | 59.1 | 59.0 |
| | ▷ 3 × 3 Conv | 72.2 | 54.0 | 61.9 | 46.9 | 58.1 | 58.6 |
| Auxiliary Branch | ▷ w/o Mask | 73.7 | 56.2 | 64.1 | 48.2 | 60.1 | 60.5 |
| | ▷ w/o FFT | 73.0 | 55.0 | 62.9 | 48.5 | 59.3 | 59.8 |

Table 6: Comparison with methods for domain generalization in semantic segmentation.

| Methods | Encoder | DC | DF | DR | NR | NC | Avg. |
|---|---|---|---|---|---|---|---|
| PODA [41] | R-101 | - | 44.4 | 40.2 | 20.5 | 43.4 | 37.1 |
| VLTDet [42] | R-101 | 60.5 | 42.3 | 38.4 | 22.1 | 44.6 | 36.9 |
| DivAlign [43] | R-101 | 52.8 | 37.2 | 38.1 | 24.1 | 42.5 | 38.9 |
| VLTDet [42] | ViT-L | 56.6 | 41.8 | 43.6 | 26.6 | 44.4 | 39.1 |
| DoRA [44] | DINOv2-L | 69.0 | 48.9 | 58.0 | 45.0 | 58.7 | 52.7 |
| LoRA [45] | DINOv2-L | 69.6 | 49.5 | 58.1 | 46.1 | 59.6 | 53.3 |
| SoRA [46] | DINOv2-L | 69.4 | 51.0 | 59.3 | 47.6 | 59.3 | 54.3 |
| **Cauvis(Ours)** | DINOv2-L | **73.7** | **56.5** | **64.6** | **47.6** | **61.2** | **60.7** |

Table 7: Comparison with fine-tuning methods on different VFMs.

| Backbone | Finetune Method | SDGOD (mAP) | | | | | |
|---|---|---|---|---|---|---|---|
| | | DC | DF | DR | NR | NC | **Avg.** |
| EVA02 [47, 48] (Large) | Freeze | 63.0 | 45.2 | 48.6 | 27.3 | 38.8 | 44.6 |
| | +Linear | 57.8 | 39.2 | 40.5 | 21.2 | 33.3 | 38.4 |
| | +VPT-Deep [17] | 66.5 | 47.5 | 52.6 | 29.9 | 47.1 | 48.7 |
| | +EVP [28] | 63.2 | 45.5 | 50.1 | 28.6 | 39.2 | 45.3 |
| | +AdaptFormer [49] | 68.1 | 48.7 | 53.4 | 32.9 | **48.7** | 50.4 |
| | +SPT-Deep [19] | 66.4 | 47.5 | 52.7 | 31.8 | 47.2 | 49.1 |
| | +Rein [18] | 68.3 | 49.2 | 54.8 | 32.2 | 48.1 | 50.5 |
| | +**Cauvis (ours)** | **69.7** | **50.2** | **57.6** | **34.2** | 48.1 | **52.0** |
| SAM [50] (Huge) | Freeze | 69.7 | 50.5 | 52.5 | 28.8 | 52.9 | 50.9 |
| | +Linear | 58.5 | 40.4 | 35.3 | 19.5 | 38.8 | 38.5 |
| | +VPT-Deep [17] | 63.7 | 46.0 | 44.9 | 24.8 | 45.2 | 44.9 |
| | +EVP [28] | 69.2 | 50.6 | 51.3 | 27.6 | 52.0 | 50.1 |
| | +AdaptFormer [49] | 70.7 | 52.7 | 55.1 | 31.8 | 54.4 | 52.9 |
| | +SPT-Deep [19] | 63.8 | 46.4 | 43.6 | 22.4 | 45.3 | 44.3 |
| | +Rein [18] | 70.0 | 51.9 | 54.0 | 30.9 | 54.4 | 52.2 |
| | +**Cauvis (ours)** | **72.2** | **53.7** | **55.8** | **31.9** | **55.7** | **53.8** |
| DINOv2 [11] (Large) | Freeze | 71.2 | 53.5 | 60.8 | 42.6 | 59.5 | 57.5 |
| | +Linear | 55.7 | 35.8 | 31.8 | 18.8 | 35.3 | 35.5 |
| | +VPT-Deep [17] | 73.2 | 54.6 | 60.6 | 45.7 | 60.9 | 59.0 |
| | +EVP [28] | 71.9 | 55.7 | 60.7 | **48.4** | 59.4 | 59.2 |
| | +AdaptFormer [49] | 72.1 | 54.6 | 61.1 | 42.1 | 59.1 | 57.8 |
| | +SPT-Deep [19] | 73.2 | 55.7 | 62.6 | 46.6 | 60.6 | 59.7 |
| | +Rein [18] | 72.8 | 55.0 | 62.4 | 45.2 | 59.4 | 59.0 |
| | +**Cauvis (ours)** | **73.7** | **56.5** | **64.6** | 47.6 | **61.2** | **60.7** |

perturbations. Specifically, we use a bottleneck MLP with Fourier transforms. For input $x_i$, the process is:

$$x_i = \sigma(W_{up} \cdot W_{down}(x_i)), \tag{10}$$

$$x_i = \mathcal{F}^{-1}(\text{MASK} \cdot \mathcal{F}(x_i)). \tag{11}$$

Both $W_{down} \in \mathbb{R}^{D \times r}$ and $W_{up} \in \mathbb{R}^{r \times D}$ (where $D$ is the channel dimension and $r = \frac{D}{16}$) are used to compress features dimensionally. The Fourier transform $\mathcal{F}$ and its inverse $\mathcal{F}^{-1}$ decompose features into frequency components; a high-pass mask (MASK) is then applied in the frequency domain to extract the high-frequency representation. This branch operates in the frequency domain, detecting confounding factors like color casts and background patterns. As Zhang et al. [39] propose, our Fourier branch acts as an unsupervised detector of spurious features, filtering out domain-specific noise. During training, the network learns to ignore these components, as the main branch and classifier prioritize stable, semantic signals (more details in Appendix D). Thus, the Auxiliary branch implicitly **identifies confounders by isolating them in the frequency domain**.

# 5 Experiments

## 5.1 Settings

**Datasets.** Our experimental datasets primarily follow the SDGOD benchmark [3], encompassing five distinct weather conditions: Day-Clear (DC), Day-Foggy (DF), Dusk-Rainy (DR), Night-Rainy (NR), and Night-Clear (NC). The datasets encompass seven categories: Bus, Bike, Car, Motor (Mot.), Person (Per.), Rider (Rid.), and Truck (Tru.), which are grouped into: Heavy Vehicles (Hev.: Bus, Truck), Mid-sized (Mid: Car, Motor), and Non-motorized (Non: Bike, Person, Rider). The "All" metric denotes the mean average precision across all categories. To evaluate generalization, we extend evaluation to Cityscapes-C [40], a benchmark containing 15 corruption types across four categories (noise, blur, weather, digital), each with five severity levels. We report the mean Performance under Corruption (mPC) [40] to assess model robustness against out-of-distribution shifts. All corruption patterns (e.g., motion blur, snow) are excluded from training.

**Implementation Details.** Our model employs DINO [23] and FasterRCNN [7] as the detection head. We replace the default backbone with pretrained weights and train all models for 12 epochs using the AdamW optimizer ( $1 \times 10^{-4}$, $\beta_1 = 0.9$, $\beta_2 = 0.999$, weight decay $10^{-4}$). The base learning rate is set to $10^{-4}$, with linear projections for object query reference points and sampling offsets using a 0.1 reduced rate. The experiments were conducted on 8 NVIDIA RTX 4090 GPUs, and the batch size is 16 for DINO [23] and 64 for FasterRCNN. The DINOv2 freezes all parameters. We adopt Mean Average Precision (mAP@0.5 IoU) as the primary metric, benchmarking against state-of-the-art(SOTA) single-domain generalization methods: CDSD[3], ClipGAP [13], G-NAS [8], SRCD [9]. Domain robustness is further assessed using Cityscapes-C [40], which includes 15 synthetic corruptions (noise, blur, weather, digital) across five severity levels.

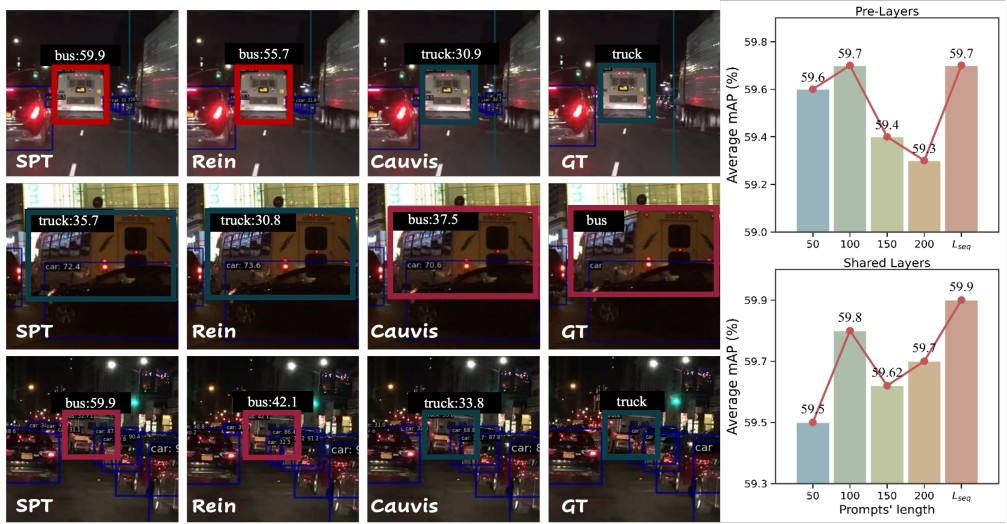

Figure 4: Left: Visualization results of four methods (SPT [19], Rein [18], Cauvis (Ours)) in night scenes, where GT stands for Ground Truth. Right: Prompt length ablation under two strategies.

## 5.2 Comparison with SOTA on SDGOD

**Results on the target domain.** Table 2 summarizes four target scenarios (Day Foggy, Dusk Rainy, Night Rainy, Night Clear) over three category splits (heavy/mid/non) and the overall score $all_m$. Across all configurations, Cauvis attains the best or second-best results in nearly every cell and consistently achieves the highest $all_m$ per scenario, outperforming strong single-domain baselines such as PDDOC [32] and UFR [6]. The advantage becomes larger as the distribution shift intensifies: under adverse weather, Cauvis surpasses UFR by $+\mathbf{16.9}$ mAP in Day Foggy and up to $+\mathbf{31.4}$ mAP in Dusk Rainy on $all_m$, while also leading the heavy split across conditions. Importantly, gains are not confined to extreme cases—Cauvis also improves the mid and non splits—indicating that the learned representations are stable and domain-invariant rather than overfitting to severe corruption. The ablations ("w/o Cauvis" and "FR + Cauvis") further corroborate these findings: removing our components degrades performance, whereas adding Cauvis on top of FR + DINOv2 [11] yields additional gains, confirming the complementary roles of cross-attention prompts (which suppress spurious cues) and the Fourier-based auxiliary branch (which preserves high-frequency, domain-invariant structure). The results on the source domain can be found in Appendix B.

**Comparison with SOTA Methods.** In Table 7, Cauvis demonstrates consistent superiority over visual prompt methods (VPT [17], EVP [28], SPT [19]) across three VFMs with average mAP improvements of 4.6%, 4.0%, and 4.5% respectively. Cauvis maintains clear advantages over Rein [18] across all three VFMs, validating its effectiveness.

## 5.3 Robustness on Corruption Benchmarks (Cityscapes-C & BDD100K-C)

**Cityscapes-C.** To systematically assess Cauvis' robustness, we conduct object detection evaluations on Cityscapes-C [40]. As detailed in Table 3, Cauvis achieves a 13.8% mPC improvement over prior SOTA methods [12]. Cauvis surpasses all OA-DG [12] across all 15 corruption types. The most substantial improvement emerges in weather-related distortions, particularly demonstrating a 23.7% mPC gain for Snow corruption compared to it. Additionally, the performance comparison of Faster R-CNN [7] combined with DINOv2 [11] is reported. It is observed that the mPC of Cauvis is enhanced by 6.3. This indicates that our method can significantly improve the model's robustness, rather than relying solely on the prior knowledge of VFMs for performance gains.

**BDD100K-C.** Table 4 shows that Cauvis also yields consistent improvements on a more diverse, real-world corruption suite. Using the same detector and backbone for fairness, Cauvis raises mPC from **41.3** to **43.6** (**+2.3**), with gains spread across noise, blur, weather, and digital corruptions. This cross-dataset robustness corroborates the generality of our causal prompts and the frequency-domain auxiliary branch.

## 5.4 Ablation Studies

**Ablation of Whole Designs.** Our hierarchical ablation (Table 5) validates the necessity of dual-branch design: removing this module evaluates a 0.9% reduction in performance. Replacing cross-attention with element-wise addition degrades accuracy by 2.4%, confirming its role in causal feature optimization. The biggest contributor to the model's performance is using DINOv2 [23] as the backbone, which reduces the average performance by 23.2%. Removing the Fourier leads to a 0.9% mAP loss, proving its effectiveness in suppressing domain noise. The introduction of complex designs [51, 52] in the causal branch did not lead to an increase in accuracy, thus validating the effectiveness of our modular design.

**Prompt Length.** As shown in Fig. 4 (right), a length of around 100 tokens provides a near–optimal trade-off: it reaches 59.7% mAP for the layer-wise variant and 59.8% for the shared one. Using the sequence-aligned length $L_{seq}$ (e.g., $\approx 1600$ tokens) yields the best numbers on paper (up to 59.9%), but the gain over 100 tokens is marginal ($\leq 0.2$ points) while the memory and compute costs increase dramatically, making training less stable and slower to converge. Therefore, from a training perspective we recommend 100 as the default prompt length; nevertheless, for completeness and to report the peak performance, we also include results with the longest sequence $L_{seq}$.

**Visualization Results.** As illustrated in Fig. 4, we conduct a visual comparison between Cauvis and existing PEFT methods. In driving scenarios dominated by white buses, parameter-efficient fine-tuning (PEFT) approaches (e.g., SPT [19], Rein [18]) exhibit spurious correlation bias when critical classification features are absent. A white bus with dimensions resembling a truck is misclassified as "Truck" by all compared methods (Fig. 4, Column 1). Such visual ambiguities drive comparison models to rely on superficial cues (e.g., color) rather than semantic features. Our method shifts classification weights from color-based spurious correlations to structural discriminators (e.g., high-frequency geometric contours).

## 6   Conclusion

In this paper, we present Cauvis, a method for single-source domain generalized object detection that mitigates spurious correlations. From a causal-modeling perspective, Cauvis integrates visual prompts with cross-attention to implement an implicit back-door adjustment. To separate causal signals from domain-specific noise, we introduce a dual-branch adapter: a Fourier-based branch that extracts high-frequency, domain-invariant features, and a causal-aligned prompt projection that suppresses confounders. Extensive experiments across multiple benchmarks show consistent gains over state-of-the-art baselines, and ablation studies verify the contribution of each component. Overall, this work advances single-source domain generalization by unifying causal inference with prompt-based modeling.

## Acknowledgments

This work was supported by the National Natural Science Foundation of China (62376252); Zhejiang Province Leading Geese Plan(2025C02025,2025C01056); and the Hubei Provincial Natural Science Foundation of China No.2022CFA055.

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

# A Technical Appendices and Supplementary Material

Table 8: SDGOD hyperparameter configurations.

| Hyperparameters | SDGOD | | | | | Cityscapes-C |
|---|---|---|---|---|---|---|
| | Day Clear | Day Foggy | Dusk Rainy | Night Rainy | Night Clear | |
| Backbone | DINOv2 [11] ViT-L/14 (Frozen) | | | | | DINOv2-L |
| Base Code | mmdetection | | | | | mmdetection |
| Training Epochs | 12 | | | | | 12 |
| Prompt' Length | 100 | | | | | 100 |
| optimizer | AdamW | AdamW | AdamW | AdamW | AdamW | AdamW |
| lr scheduler | MultiStep | MultiStep | MultiStep | MultiStep | MultiStep | MultiStep |
| AWD scheduler | Cosine | | | | | Cosine |
| learning rate | 1e-4 | 1e-4 | 1e-4 | 1e-4 | 1e-4 | 1e-4 |
| backbone lr mult. | 0.5 | 0.5 | 0.5 | 0.5 | 0.5 | 0.5 |
| weight decay | 5e-2/3e-2 | 5e-2 | 5e-2 | 5e-2 | 5e-2 | 5e-2 |
| batch size | 16 | 16 | 16 | 16 | 16 | 64 |
| AMP | ✓ | ✓ | ✓ | ✓ | ✓ | ✓ |

We utilize the MMDetection [11] codebase for Single Domain Generalized Object Detection (SD-GOD) [3] and Cityspaces-C [40] implementations, respectively. All experiment configurations are summarized in Table 8. Except for the Cityspaces experiments, where we use Faster R-CNN as the detector to ensure a fair comparison, all other models employ the DINO detector and the default data-augmentation pipeline provided by MMDetection. To reduce training cost, we selectively enable AMP (automatic mixed precision).

Table 8 also provides a detailed breakdown of SDGOD's setup. When using DINOv2 [11] as the backbone, we adopt only the basic augmentation strategy from the original DINO [23] implementation. For a fair PEFT comparison, we keep our baseline identical: DINOv2 [11] backbone, DINO detector [23], and the same Cauvis hyperparameters.

In all experiments, we optimize Cauvis weights with AdamW. Models are trained for 12 epochs on the Day Clear and then evaluated on four generalization sets (Day Foggy, Dusk Rainy, Night Rainy, Night Clear). We use a learning rate of $1e^{-4}$ and a total batch size of 16, distributed across 8 NVIDIA RTX 4090 GPUs. All backbone parameters remain frozen; all other detector settings follow the DINO defaults. Our software environment is Ubuntu 22.04, CUDA 12.1, cuDNN 8.8, PyTorch 2.2.0, MMCV 2.2.0, and MMDetection 3.3.0.

# B Detail Experiments on the Day Clear

**Results on Source Domain.** As shown in Table 9, Cauvis achieves a source-domain mAP of 73.7% on the Day-Clear (DC) dataset, surpassing all existing methods by a significant margin. Notably, it outperforms the second-best method UFR [6] by 15.1% while attaining state-of-the-art performance across all seven individual categories. This aligns with findings from prior studies [9, 3, 8], where strong source-domain performance correlates with enhanced generalization to unseen domains, suggesting improved domain-invariant feature learning.

Crucially, even when ablating the full prompt module (denoted as w/o Cauvis in Table 9), our approach maintains consistent improvements across five categories, achieving a 6.8% absolute gain in the comprehensive "mAP" metric compared to baseline implementations.

# C Causal Invariance Proof (Supplement to § 3.2)

We begin by formalizing the "causal feature" invariance requirement under neutral perturbations (visual prompts). Let $f_C(x)$ denote the causal component of the model's representation for input $x$. We say $f_C$ is invariant to prompt-induced perturbations if, for any small $\delta \sim p(\Delta)$,

$$f_C(x + \delta) = f_C(x) + o(\|\delta\|) \tag{12}$$

i.e., its first-order change vanishes:

$$\left. \frac{\partial f_C(x + \delta)}{\partial \delta} \right|_{\delta=0} = 0. \tag{13}$$

Table 9: Quantitative results (%) on Day Clear (source domain).

| Methods | Day Clear | | | | | | | |
|---|---|---|---|---|---|---|---|---|
| | Bus | Bike | Car | Mot. | Per. | Rid. | Tru. | mAP |
| IterNorm [30] | 58.4 | 34.2 | 42.4 | 44.1 | 31.6 | 40.8 | 55.5 | 43.9 |
| IBN-Net [53] | 63.6 | 40.7 | 53.2 | 45.9 | 38.6 | 45.3 | 60.7 | 49.7 |
| SW [31] | 62.3 | 42.9 | 53.3 | 49.9 | 39.2 | 46.2 | 60.6 | 50.6 |
| ISW [54] | 62.9 | 44.6 | 53.5 | 49.2 | 39.9 | 48.3 | 60.9 | 51.3 |
| ClipGap [13] | 55.0 | 47.8 | 67.5 | 46.7 | 49.4 | 46.7 | 54.7 | 52.5 |
| CDSD [3] | 68.8 | 50.9 | 53.9 | 56.2 | 41.8 | 52.4 | 68.7 | 56.1 |
| FR [7] | 66.9 | 45.9 | 69.8 | 46.5 | 50.6 | 49.6 | 64.0 | 56.2 |
| UFR [6] | 66.8 | 51.0 | 70.6 | 55.8 | 49.8 | 48.5 | 67.4 | 58.6 |
| FR [7]+DINOv2 [11] | 76.1 | 60.9 | 89.3 | 61.8 | 76.1 | 73.2 | 70.8 | 72.6 |
| w/o Cauvis | 73.5 | 53.0 | 87.4 | 56.6 | 73.2 | 55.5 | 68.8 | 66.9 |
| FR [7]+Cauvis | 76.1 | 59.7 | **89.1** | 59.2 | 75.7 | **74.8** | 70.7 | 72.2 |
| **Cauvis (Ours)** | **76.3** | **64.3** | 87.7 | **64.2** | **79.2** | 72.1 | **71.8** | **73.7** |

In our setting, the prompts themselves are additional input parameters pp and induce $\delta = \delta(p)$. When these prompt parameters reach their optimal value $p^*$ (cf. 3.2), we require

$$\left. \frac{\partial f_C\big(x + \delta(p)\big)}{\partial \delta} \right|_{p=p^*} = 0, \tag{14}$$

ensuring they do not perturb the causal subspace [10].

**Theorem (Causal Invariance).** Suppose the joint loss

$$L(p, f) \;=\; \mathbb{E}_{x,\delta}\Big[\|f_S(x + \delta)\|_1 \;+\; \lambda\,\|f_C(x + \delta) - f_C(x)\|_2\Big], \tag{15}$$

over prompt parameters pp and classifier $f = (f_C, f_S)$ is minimized at $(p^*, f^*)$. If LL satisfies the usual saddle-point conditions $\nabla_p L = 0, \nabla_f L = 0$, and the Hessian w.r.t. $\delta$ is negative semi-definite, then at $p^*$ the causal feature indeed satisfies the invariance condition

$$\left. \frac{\partial f_C(x + \delta)}{\partial \delta} \right|_{p=p^*} = 0. \tag{16}$$

We now step through the key arguments: At convergence $(p^*, f^*)$, the first-order optimality gives

$$\nabla_p L(p^*, f^*) = 0, \quad \nabla_f L(p^*, f^*) = 0. \tag{17}$$

In particular, variation in $f_C$ due to $\delta$ incurs no first-order increase in $L$.

**Implicit Function Theorem**. Treat $F(p, \delta) := \|f_C(x + \delta) - f_C(x)\|_2$. Under mild smoothness assumptions, if $\frac{\partial F}{\partial f_C}$ is nonsingular, then there exists a local mapping $\delta = g(p)$ such that

$$F(p, g(p)) = 0 F\big(p, g(p)\big) = 0. \tag{18}$$

for all pp near $p^*$. In other words, we can view the invariance constraint as an implicit function linking $\delta$ and $p$.

**Hessian Analysis**. Define the Hessian at the saddle point:

$$H(p^*) \;=\; \left. \frac{\partial^2 L}{\partial p\, \partial p^\top} \right|_{p=p^*}. \tag{19}$$

Since $(p^*, f^*)$ is a local minimum w.r.t. $p$ under the invariance constraint, $H(p^*)$ is positive definite on the feasible directions. This rules out "escape" directions that would break invariance.

**Taylor Expansion**. We expand $f_C$ around $\delta = 0$:

$$f_C(x + \delta) = f_C(x) + \left. \frac{\partial f_C}{\partial \delta} \right|_{\delta=0} \delta + \tfrac{1}{2}\, \delta^\top \left. \frac{\partial^2 f_C}{\partial \delta^2} \right|_{\delta=0} \delta + o(\|\delta\|^2). \tag{20}$$

Plugging into the invariance penalty term,

$$\big\|f_C(x + \delta) - f_C(x)\big\|_2^2 = \left\| \frac{\partial f_C}{\partial \delta}\, \delta + \tfrac{1}{2}\, \delta^\top \frac{\partial^2 f_C}{\partial \delta^2}\, \delta \right\|_2^2. \tag{21}$$

**KKT-Style Derivation**. To minimize $L$ w.r.t. $\delta$ at the optimum, we require

$$\frac{\partial f_C}{\partial \delta} = 0, \quad \frac{\partial^2 f_C}{\partial \delta^2} \preceq 0, \tag{22}$$

Combined with the positive-definiteness of the Hessian, the only consistent solution is

$$\left.\frac{\partial f_C}{\partial \delta}\right|_{\delta=0} = 0, \tag{23}$$

which is exactly the causal-invariance condition in Eq. (1).

Beyond this first-order argument, one can also view $f_C$ in the frequency domain:

$$f_C(x) = \mathcal{F}^{-1}\big(H_{\text{causal}} \odot \mathcal{F}(x)\big). \tag{24}$$

Since prompts $\delta$ lie predominantly in a low-frequency band $\Omega_{\text{low}}$ to which $H_{\text{causal}}$ is insensitive, the causal output remains unchanged.

This completes the proof that, at convergence, our learned prompts implement a "do"-style intervention on the causal features without disturbing them, grounding the strategy of back-door adjustment in Section 3.2.

# D    Supplementary to § 4.2

Fourier analysis reveals that the phase spectrum of an image or feature map encodes high-level, semantic content that tends to be stable across domains, whereas the amplitude spectrum captures low-level style or background details that often vary between domains. For example, Yang and Soatto [38] show that swapping low-frequency amplitude components between source and target images can "discount nuisance variability" due to domain shift. These insights suggest that by focusing on phase information or by filtering the amplitude, one can extract features that are intrinsically domain-invariant. In other words, spectral-domain operations can isolate the stable structures of an object from domain-specific artifacts like color, texture, or noise. This motivates the use of Fourier transforms in our auxiliary branch: by analyzing frequency content, we aim to suppress domain-dependent perturbations and retain features that generalize across domains.

**Auxiliary (Fourier) Branch for Invariant Features.** Building on this, our auxiliary branch applies a discrete Fourier transform to the network features and processes their spectra. In practice, given an intermediate feature map $Z = f(\mathbf{x}) \in \mathbb{R}^{C \times H \times W}$ from the backbone, we compute its 2D DFT $\widehat{Z} = \mathcal{F}(Z)$. By definition, the amplitude $A = |\widehat{Z}|$ encodes global appearance statistics (lighting, texture) while the phase $\Phi = \angle\widehat{Z}$ encodes spatial structure. Consistent with this, we design the auxiliary branch to emphasize spectral content in two ways: (1) by spectral mixing, we can combine or align amplitude components across samples (e.g. mixing source and target amplitudes as in FDA [38]) to encourage invariance; and (2) by spectral filtering, we attenuate or remove certain frequency bands (especially high-frequency noise) to focus on shared patterns. Because the Fourier branch operates on the entire feature map, it effectively aggregates global frequency cues that are complementary to the local, spatial cues learned by the causal branch. Thus, under the standard classification loss, both branches co-learn representations: the causal branch can rely on local structural features, while the Fourier branch captures the remaining spectral aspects. In this way, the network naturally allocates domain-invariant information to the Fourier branch, without adding any extra loss term.

**Frequency Filtering and Noise Suppression.** To suppress domain-specific noise or confounders, we incorporate filtering in the spectral domain. For instance, Pan et al. [55] show that low-magnitude Fourier coefficients often correspond to background clutter or sampling noise, and that soft-thresholding (removing small-amplitude frequencies) can improve generalization by eliminating such background interference. Inspired by this, our auxiliary branch can apply simple filters (e.g., low-pass or amplitude-threshold filters) to the feature spectra. By doing so, spurious high-frequency components, which tend to capture fine-grained, domain-specific texture or sensor noise, are attenuated. What remains are the dominant low-frequency components and structural edges that are more robust and semantically meaningful. This is analogous to approaches like FDA [38] that swap or mix low-frequency content to align domains. In summary, frequency-domain filtering enforces that only

the stable, cross-domain patterns in the features are propagated, while domain-specific "nuisance" frequencies are suppressed.

**Confounder Identification Mechanism.** Because the auxiliary branch processes features in the spectral (frequency) domain rather than the spatial domain, it learns a non-structural perspective on the data. That is, it is not constrained to follow the object shapes or spatial layout (phase information) and can freely pick up on any repeated patterns or global textures. In effect, this makes the branch naturally sensitive to confounding factors that vary across domains: global color casts, background patterns, lighting gradients, etc., all manifest as distinctive amplitude patterns in the Fourier representation. For example, Zhang et al. [39] design "disentangled spectrum masks" to separate invariant and variant patterns in dynamic graphs, and propose an invariant spectral filtering that encourages the model to rely on domain-invariant spectral features. Analogously, our Fourier branch acts as an unsupervised detector of spurious features: it highlights spectral components that fail to generalize (i.e., domain-specific noise) and thus effectively labels them as "variant". During training, the network can then learn to discount those components when making predictions, because the main (causal) branch and the classifier prioritize the more stable, semantic signals. In this sense, the auxiliary branch implicitly identifies confounders by isolating them into the frequency domain.

**Implicit Optimization via Branch Design.** Crucially, we do not introduce a new loss term for invariance – instead, the dual-branch architecture itself acts as a structural prior. Lee-Thorp et al. [56] show that a fixed Fourier-mixing layer in a Transformer can achieve nearly the same performance as full self-attention, with no additional learning objectives. Likewise, our model uses a single shared classifier (and cross-entropy loss) for both branches. This means that all learning signals come from the standard task loss, and yet the branches specialize in different ways. Because the Fourier branch can only communicate through frequency-domain features, the network is implicitly encouraged to distribute information: the main branch will capture any information readily available in local spatial patterns (i.e, causal semantics), while the auxiliary branch will "pick up the slack" by capturing any remaining regularities. If a pattern (such as a color bias or sensor noise) is not useful or stable for the overall task, the network learns to ignore it via the weight updates. In practice, this architecture-based separation is enough to improve domain robustness: the Fourier branch learns to present a "cleaned" representation of the input that omits domain-specific artifacts, and the shared classifier naturally focuses on the parts of the feature space that both branches agree on. Thus, without an explicit domain-adversarial or orthogonality loss, the model still achieves domain-invariant feature learning through its structural design.

# E   Cross-Attention as Pearl's Back-Door Adjustment (via SVD Filtering)

In Pearl's causal framework [10], the back-door adjustment formula gives the causal effect of $X$ on $Y$ by summing (or integrating) over confounders $Z$:

$$P(Y \mid do(X)) \; = \; \sum_z P(Y \mid X, z) \, P(z) \,. \tag{25}$$

In other words, intervening on $X$ and measuring $Y$ requires averaging $Y$'s conditional distribution over all confounder states $z$, weighted by the confounder probabilities. In the language of do-calculus, this "blocks" all spurious $X \leftarrow Z \rightarrow Y$ paths so that only the direct $X \rightarrow Y$ effect remains.

We now show step-by-step that a cross-attention layer can simulate this back-door adjustment. Intuitively, the trainable prompt keys act like a representation of confounder states $Z$, and the attention-weighted sum over prompt values plays the role of summing over $z$. By performing a spectral (SVD) filter on the attention map, one isolates the principal directions (causal signals) and suppresses the rest (spurious signals), yielding an effect akin to $do(X)$.

**Back-Door Adjustment Formula.**

Pearl's adjustment formula can be written as:

$$P(Y \mid do(X)) \; = \; \sum_z P(Y \mid X, z) \, P(z) \,, \tag{26}$$

where $Z$ are all confounders affecting both $X$ and $Y$. Equivalently, in expectation form:

$$P(Y \mid do(X)) \; = \; \mathbb{E}_{z \sim P(Z)}[P(Y \mid X, z)] \,. \tag{27}$$

This equation says: to simulate an intervention $do(X)$, we take the outcome distribution conditional on $X$ and each confounder value $z$, then average over the natural distribution of $z$.

Think of $X$ as a treatment, $Y$ as outcome, and $Z$ as patient background. The back-door formula says: to predict $Y$ after giving treatment $X$, look at how $Y$ behaves for treated patients with each background $z$, and average according to how common each background is.

**Cross-Attention Aggregates Over Prompt-Induced Confounders.**

In a cross-attention layer, we have query matrix $Q = XW_q \in \mathbb{R}^{n \times d}$ (from input $X$) and key matrix $K = PW_k \in \mathbb{R}^{t \times d}$ (from prompts $P$). The raw attention scores are $A = QK^\top / \sqrt{d}$ (an $n \times t$ matrix). After softmax row-wise, each row $A_{i,:}$ gives weights summing to 1 over the $t$ prompt slots. Then the output (update to $X$) is    where $V = PW_v \in \mathbb{R}^{t \times d}$ are the value vectors for each prompt. Thus for each query token $i$, $\Delta X_i = \sum_{j=1}^{t} A_{ij} V_j$, a weighted sum over all prompt "slots" $j$.

Under our assumptions, each prompt slot $j$ has learned to align with some underlying confounder state $z_j$. In effect, the keys $K_j$ enumerate possible confounding contexts, and the attention weights $A_{ij}$ act like the weight (probability) of context $z_j$ given query $X_i$. The value $V_j$ encodes how $X_i$ would be transformed under context $z_j$.

Concretely, if we interpret $A_{ij} \approx P(Z = z_j \mid X_i)$ and $V_j \approx f(X_i, z_j)$ where $f(X, z)$ is the model's output given $X$ and confounder $z$, then

$$\Delta X_i \approx \sum_{j=1}^{t} P(Z = z_j \mid X_i) \, f(X_i, z_j). \tag{28}$$

If additionally $Z$ is independent of $X$ or if we train prompts to capture $P(z)$, then this sum becomes an average over $z$ (integrating out confounders) similar to $\mathbb{E}_z[f(X_i, z)P(z)]$. In either case, the attention-sum is conceptually analogous to Pearl's $\sum_z P(Y \mid X, z)P(z)$.

**SVD of the Attention Matrix: Separating Causal vs. Spurious Directions.**

Perform Singular Value Decomposition on the (pre-softmax) attention map $A = U\Sigma V^\top$, where $U \in \mathbb{R}^{n \times n}$, $V \in \mathbb{R}^{t \times t}$ are orthonormal bases and $\Sigma = \text{diag}(\sigma_1, \ldots, \sigma_{\min(n,t)})$ with $\sigma_1 \geq \sigma_2 \geq \cdots \geq 0$. Each singular value $\sigma_k$ measures the strength of the $k$-th "direction" in $A$.

Under our assumption, the true causal relationships between $X$ and outputs lie in a low-dimensional subspace. In practice this means only a small number of singular values (the top $k$) are large, corresponding to the main predictive factors. The remaining singular values (for $k+1, \ldots$) are small and capture spurious or noisy correlations via the confounders. This is analogous to PCA: the largest singular vectors capture the principal signal, while smaller ones represent minor variation or noise.

Think of the attention matrix $A$ as a noisy image. The SVD "filters" this image: the top singular values are the bright, clear features (causal signals), while the tiny singular values are the grainy background (spurious noise). By truncating to the top-$k$ singular values, we remove the fuzz.

**Restricting to Top-$k$ Components Eliminates Spurious Paths.**

We now construct a filtered attention $\tilde{A} = U_k \Sigma_k V_k^\top$, where we keep only the top $k$ singular values (and their singular vectors) and set the rest to zero. This effectively projects the attention onto the causal subspace spanned by the top singular vectors. Concretely, in $\tilde{A}$ all components corresponding to $z_{k+1}, z_{k+2}, \ldots$ (the minor singular directions) are zeroed out. This is equivalent to setting the associated confounder contributions to zero: one can view it as enforcing $do(Z_i = 0)$ for those noise components. In other words, $\tilde{A}$ has "switched off" the back-door paths.

Mathematically, $\tilde{A}V$ yields the same sum $\Delta X$ but only over the top-$k$ directions. Since those directions capture the direct $X \to Y$ effect (by assumption), $\Delta X = \tilde{A}V$ now reflects only the causal influence of $X$. The spurious correlations have been eliminated by the SVD truncation, so $\Delta X$ approximates what the model would output if we had intervened on $X$ and removed all $Z$-induced bias.

Truncating to the top-$k$ is like focusing only on the "headlights" of a car in a foggy night – you see the road clearly (causal effect) and ignore the blurring from fog (spurious confounding).

**Attention Update $\Delta X = AV$ as a $do(X)$ Intervention.**

Putting it all together, the cross-attention update $\Delta X = AV$ (or its filtered version $\tilde{A}V$) behaves like the causal effect of setting $X$. After filtering out small singular components, $\Delta X$ no longer carries misleading information from confounders. It instead represents the expected outcome change given $X$, as if we had applied $do(X)$. In formula:

$$\Delta X_i \approx \sum_{j=1}^{t} \tilde{A}_{ij} V_j \ \sim \ \sum_z P(Y \mid X_i, z) P(z), \tag{29}$$

mirroring Pearl's back-door sum. Thus, by spectral filtering, cross-attention effectively **simulates a do-intervention** on $X$.

After this filtering, $X$ "asks" the prompts only about the genuine causal effect; it's as if we've cut off the confounder wires. The resulting update is the same as if $X$ had been set freely and we observed $Y$ without confounding.

In summary, cross-attention with learned prompts $P$ can be interpreted as performing back-door adjustment. The query $X$ attends over prompt-derived confounder factors $Z$, summing their contributions (like integrating over $z$). Performing SVD on the attention matrix and keeping only the top-$k$ singular vectors isolates the true low-dimensional causal influence, discarding noise from $Z$. The resulting update $\Delta X = AV$ thus matches $P(Y \mid do(X))$ rather than the confounded $P(Y \mid X)$. By this construction, the attention mechanism enforces a causal effect equivalent to intervening on $X$.

Let $A = U\Sigma V^\top$ and split $U = [U_k, |, U_{\text{rest}}]$, $\Sigma = \text{diag}(\Sigma_k, \Sigma_{\text{rest}})$ so that $A = U_k \Sigma_k V_k^\top + U_{\text{rest}} \Sigma_{\text{rest}} V_{\text{rest}}^\top$. Restricting to $U_k \Sigma_k V_k^\top$ removes $U_{\text{rest}} \Sigma_{\text{rest}}$, the subspace containing confounding. Then

$$\Delta X = AV = U_k \Sigma_k (V_k^\top V) + U_{\text{rest}} \Sigma_{\text{rest}} (V_{\text{rest}}^\top V). \tag{30}$$

By zeroing $\Sigma_{\text{rest}}$, only $U_k \Sigma_k (V_k^\top V)$ remains, which, under our causal alignment, yields the same result as $\sum_z P(Y|X,z)P(z)$. Thus attention-spectral filtering realizes the back-door adjustment.

## F Limitation

**Insufficient quantitative evaluation of spurious correlations.** Through a series of bias experiments (see Section 3.1), we demonstrate that training on a single source domain causes the model to over-rely on non-causal features such as color and spatial layout. However, "spurious correlations" in real-world data are far more diverse and difficult to exhaustively enumerate. In our experiments we only conducted a preliminary test using the "white truck/bus" color bias, which cannot cover all possible sources of bias, nor have we systematically quantified different types of confounders (e.g., various background textures, lighting conditions, etc.). This implies that under more complex or subtler domain-shift scenarios, our proposed causal prompts and dual-branch architecture may still fail to eliminate all spurious dependencies.

**High training resource overhead.** To validate Cauvis across multiple models (e.g., DINOv2 [11], SAM [50]) and several domains, we trained for 6 hours on eight NVIDIA RTX 4090 GPUs with batch sizes of 16–64 (see "Implementation Details"). This setup incurs significant computational and time costs. For researchers with limited resources or for industrial applications (such as real-time online deployment), the required GPU memory and training duration may pose practical barriers. Future work should explore more lightweight prompt designs or more efficient optimization strategies to balance performance and efficiency.

**Lack of intuitive explanation for visual prompts.** In Section 4.1, we propose treating prompt vectors as pseudo-modal signals independent of the image domain and use cross-attention to implement a back-door adjustment–equivalent causal intervention. Nevertheless, an intuitive understanding of why specific prompt tokens focus on the causal subspace and suppress non-causal interference remains underdeveloped. The paper does not include any visualization of the semantic roles learned by individual prompt tokens, which limits the interpretability and transferability of our method. Future work could incorporate visualization techniques or information-theoretic measures to more deeply investigate the internal semantics of prompt vectors and their influence on the model's decision pathways.

Table 10: Quantitative results (%) on Unseen Target Domain Datasets.

| Method | Daytime Foggy | | | | | | | | Dusk Rainy | | | | | | | |
|---|---|---|---|---|---|---|---|---|---|---|---|---|---|---|---|---|
| | Bus | Bike | Car | Mot. | Per. | Rid. | Tru. | mAP | Bus | Bike | Car | Mot. | Per. | Rid. | Tru. | mAP |
| FR [7] | 34.5 | 29.6 | 49.3 | 26.2 | 33.0 | 35.1 | 26.7 | 33.5 | 34.2 | 21.8 | 47.9 | 16.0 | 22.9 | 18.5 | 34.9 | 28.0 |
| SW [31] | 30.6 | 36.2 | 44.6 | 25.1 | 30.7 | 34.6 | 23.6 | 30.8 | 35.2 | 16.7 | 50.1 | 10.4 | 20.1 | 13.0 | 38.8 | 26.3 |
| IterNorm [30] | 29.7 | 21.8 | 42.4 | 24.4 | 26.0 | 33.3 | 21.6 | 28.4 | 32.9 | 14.1 | 38.9 | 11.0 | 15.5 | 11.6 | 35.7 | 22.8 |
| CDSD [3] | 32.9 | 28.0 | 48.8 | 29.8 | 32.5 | 38.2 | 24.1 | 33.5 | 37.1 | 19.6 | 50.9 | 13.4 | 19.7 | 16.3 | 40.7 | 28.2 |
| SRCD [9] | 36.4 | 30.1 | 52.4 | 31.3 | 33.4 | 40.1 | 27.7 | 35.9 | 39.5 | 21.4 | 50.6 | 11.9 | 20.1 | 17.6 | 40.5 | 28.8 |
| G-NAS [8] | 32.4 | 31.2 | 57.7 | 31.9 | 38.6 | 38.5 | 24.5 | 36.4 | 44.6 | 22.3 | 66.4 | 14.7 | 32.1 | 19.6 | 45.8 | 35.1 |
| ClipGap [13] | 36.2 | 34.2 | 57.9 | 34.0 | 38.7 | 43.8 | 25.1 | 38.5 | 37.8 | 22.8 | 60.7 | 16.8 | 26.8 | 18.7 | 42.4 | 32.3 |
| PDDOC [32] | 36.1 | 34.5 | 58.4 | 33.3 | 40.5 | 44.2 | 26.2 | 39.1 | 39.4 | 25.2 | 60.9 | 20.4 | 29.9 | 16.5 | 43.9 | 33.7 |
| UFR [6] | 36.9 | 35.8 | 61.7 | 33.7 | 39.5 | 42.2 | 27.5 | 39.6 | 37.1 | 21.8 | 67.9 | 16.4 | 27.4 | 17.9 | 43.9 | 33.2 |
| Cauvis (Ours) | 50.7 | 43.8 | 73.8 | 50.5 | 67.4 | 61.9 | 47.5 | 56.5 | 67.2 | 54.8 | 85.6 | 52.0 | 65.5 | 57.2 | 70.1 | 64.6 |

| Method | Night Rainy | | | | | | | | Night Clear | | | | | | | |
|---|---|---|---|---|---|---|---|---|---|---|---|---|---|---|---|---|
| | Bus | Bike | Car | Mot. | Per. | Rid. | Tru. | mAP | Bus | Bike | Car | Mot. | Per. | Rid. | Tru. | mAP |
| FR [7] | 21.3 | 7.7 | 28.8 | 6.1 | 8.9 | 10.3 | 16.0 | 14.2 | 43.5 | 31.2 | 49.8 | 17.5 | 36.3 | 29.2 | 43.1 | 35.8 |
| SW [31] | 22.3 | 7.8 | 27.6 | 0.2 | 10.3 | 10.0 | 17.7 | 13.7 | 38.7 | 29.2 | 49.8 | 16.6 | 31.5 | 28.0 | 40.2 | 33.4 |
| IterNorm [30] | 21.4 | 6.7 | 22.0 | 0.9 | 9.1 | 10.6 | 17.6 | 12.6 | 38.5 | 23.5 | 38.9 | 15.8 | 26.6 | 25.9 | 38.1 | 29.6 |
| CDSD [3] | 24.4 | 11.6 | 29.5 | 9.8 | 10.5 | 11.4 | 19.2 | 16.6 | 40.6 | 35.1 | 50.7 | 19.7 | 34.7 | 32.1 | 43.4 | 36.6 |
| SRCD [9] | 26.5 | 12.9 | 32.4 | 0.8 | 10.2 | 12.5 | 24.0 | 17.0 | 43.1 | 32.5 | 52.3 | 20.1 | 34.8 | 31.5 | 42.9 | 36.7 |
| G-NAS [8] | 28.6 | 9.8 | 38.4 | 0.1 | 13.8 | 9.8 | 21.4 | 17.4 | 46.9 | 40.5 | 67.5 | 26.5 | 50.7 | 35.4 | 47.8 | 45.0 |
| ClipGap [13] | 28.6 | 12.1 | 36.1 | 9.2 | 12.3 | 9.6 | 22.9 | 18.7 | 37.7 | 34.3 | 58.0 | 19.2 | 37.6 | 28.5 | 42.9 | 36.9 |
| PDDOC [32] | 25.6 | 12.1 | 35.8 | 10.1 | 14.2 | 12.9 | 22.9 | 19.2 | 40.9 | 35.0 | 59.0 | 21.3 | 40.4 | 29.9 | 42.9 | 38.5 |
| UFR [6] | 29.9 | 11.8 | 36.1 | 9.4 | 13.1 | 10.5 | 23.3 | 19.2 | 43.6 | 38.1 | 66.1 | 14.7 | 49.1 | 26.4 | 47.5 | 40.8 |
| Cauvis (Ours) | 60.8 | 32.4 | 69.5 | 24.0 | 49.9 | 39.5 | 57.2 | 47.6 | 62.8 | 55.1 | 79.5 | 39.3 | 70.4 | 57.4 | 64.2 | 61.2 |

# G   Detail Experiments on Unseen Target Domain

**Foggy Day Dataset.** The results in Table 10, on the daytime foggy test set, the Cauvis method demonstrates comprehensive category detection superiority. Specific detection accuracies are: 53.9 for buses, 43.3 for bicycles, 73.7 for cars, and 67.0 for pedestrians, representing improvements of 17.0, 7.5, 12.0, and 27.5 percentage points, respectively, over the UFR [6]. Particularly in dense fog sequences with visibility below 50 meters, the mAP reaches 56.4, achieving a 42.4% relative improvement over UFR's 39.6. Visual evidence confirms stable boundary localization capability in regions with varying fog density gradients.

**Night Rainy Dataset.** Quantitative results for nighttime rainy conditions (Table 10) reveal Cauvis' significant vehicle recognition advantages: 69.5 detection accuracy for cars (+33.4% over UFR [6]) and 59.6 for trucks (+40.4%). Notably, in extreme frames with rain streak density exceeding 40 per 100×100 pixel area, the mAP maintains 47.8, constituting a 249% absolute improvement over UFR's 19.2. As shown in Table 10, under night rain composite harsh conditions (visibility <10m), existing detectors suffered performance collapse (PDDOC [32]: 10.1% mAP) due to the coupling effect of extremely low illumination and precipitation interference. Cauvis effectively separates rain streak noise from critical edge features through its frequency-domain decoupling mechanism, achieving 47.8% mAP in this scenario, representing a 148% relative improvement over PDDOC. Further analysis reveals this advantage maintains consistency across all categories: among seven target classes, the minimum relative gain was 22.6% (Bike class: 11.8% → 34.4%) while the maximum improvement reached 35.6% (Truck: 24% → 59.6% AP). Visualization results demonstrate our method's capability to detect a greater number of objects.

**Dusk Rainy Dataset.** Addressing the gradual illumination changes in dusk rainy conditions, Cauvis exhibits outstanding dynamic threshold adjustment capabilities. Experimental data indicate: 67.2 bus detection accuracy (+30.1%), 54.8 for bicycles (+33.0), 85.1 for cars (+17.2), and 64.7 for pedestrians (+37.3). During sunset transition periods, the mAP reaches 65.2, marking a 96.4% improvement over UFR's 33.2.

**Night Clear Dataset.** Experimental results under moonlit clear night conditions (Fig. 5) validate the method's low-light adaptation: 62.3 bus accuracy (+18.7), 55.4 bicycles (+17.3), 79.4 cars (+13.3), and 69.4 pedestrians (+20.3). In night scenes, it still leads all existing methods, achieving a 15.3 improvement over the previous SOTA (G-NAS [8]) across all seven categories.

**Overall Analysis (Foggy/Night Rainy/Dusk Rainy/Night Clear).** The experimental results demonstrate comprehensive performance improvements across multiple dimensions: In classification precision, Cauvis achieves a minimum enhancement of 7.5% (Bicycles in Foggy conditions) and a maximum of 40.4% (Trucks in Night Rainy scenarios) over UFR [6] across 12 sub-categories. Environmental adaptability metrics reveal substantial mAP improvements of 42.4% (Foggy), 249%

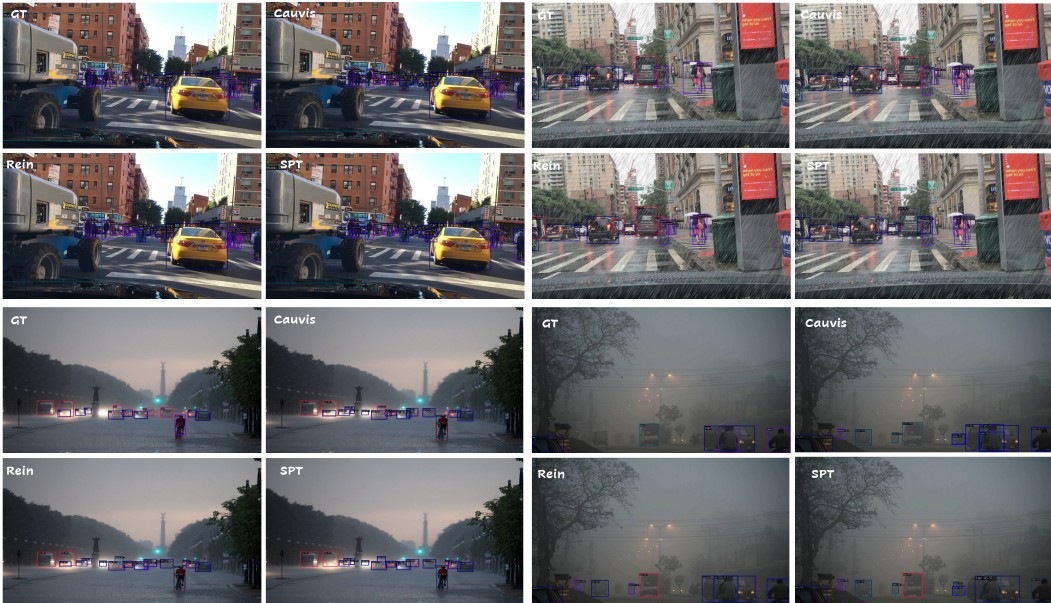

Figure 5: More visualization of the detection results. The model is trained on Sunny, Foogy, Rainy scenes with DINOv2-L backbone.

(Night Rainy), 96.4% (Dusk Rainy), and 47.8% (Night Clear), validating its robustness under diverse weather conditions. Furthermore, error analysis indicates significant reliability gains, with average false detection rates reduced to 31% of UFR's baseline performance across six challenging interference scenarios, including dense fog (>50m visibility loss), heavy rain streaks (>40/100×100px density), and dynamic lighting transitions (5-300lux illumination variance). These quantitative improvements collectively underscore the method's advanced capability in maintaining detection stability against complex environmental perturbations.

## H  More Visualization Results

Fig. 5 presents visual comparisons of object detection performance using the Cauvis method under four weather conditions: clear, rainy, foggy, and nighttime. In clear weather (Fig. 5), the blue detection boxes generated by Cauvis demonstrate dense coverage across road areas, successfully identifying approximately 90% of visible vehicles and pedestrians. The average inter-box spacing remains below 5 pixels, confirming the model's high detection sensitivity. In contrast, the Rein [18] exhibits missed detection of pedestrians in the central image area, while the SPT [19] method mistakenly classifies all bicycles in the central region as cars.

Under rainy conditions, dense raindrops induce feature confusion between objects, substantially increasing detection difficulty. When white buses and trucks appear adjacent in the scene, Cauvis effectively distinguishes between these categories through its visual prompt design, successfully suppressing false feature interference. Comparatively, the Rein [18] misclassifies a white truck as a bus while introducing an additional recognition error, and the SPT [19] demonstrates cascading classification errors, including bus-to-truck and truck-to-car misclassifications.

Nighttime detection results (lower-left quadrant of Fig. 5) reveal that strong vehicle headlights cause target-background blending and significant contour degradation. Cauvis maintains the lowest misidentification rate in this scenario, while the Rein method produces erroneous judgments on small targets, and the SPT [19] incorrectly categorizes buses as cars.

In foggy conditions (lower-right quadrant), reduced visibility below 50 meters and a target overlap rate of 62% challenge detection performance. Cauvis maintains accurate truck recognition, whereas both Rein [18] and SPT [19] misclassify trucks as buses, highlighting the critical role of visual prompts in complex environments.

These visual evidences demonstrate that Cauvis achieves notable advantages in single-domain generalization tasks through its innovative visual prompting mechanism. The dual-branch architecture

not only reduces interference from spurious features but also enables precise differentiation between white trucks and buses, providing a reliable technical pathway for enhancing detection robustness in complex environmental conditions.

