# OpenReview forum: "Towards Single-Source Domain Generalized Object Detection via Causal Visual Prompts"
_NeurIPS.cc/2025/Conference — NeurIPS 2025 poster_

### Official Review · Reviewer_pbD8 · 2025-06-24

**Clarity:** 3
**Significance:** 2
**Originality:** 2
**Rating:** 4
**Confidence:** 5

**Summary:**

This paper focuses on Single-Domain Generalized Object Detection, training an detector on an annotated source domain and generalizing it to unseen domains. To increase the generalization ability, this paper proposes a cross-attention prompts to learn visual domain-invariant knowledge and a dual-branch adapter to disentangle causal-spurious features. Experiments verify the effectiveness of the proposed method.

**Questions:**

Please refer to Weakness.

**Ethical Concerns:**

["NO or VERY MINOR ethics concerns only"]

**Final Justification:**

Thanks for the author's' detailed rebuttal. Most of my concerns have been addressed, so I maintain my original positive score.

However, I still believe that the introduction of DINOv2 into SDGOD cannot be considered a contribution. While the DA task and the DG task have different priors, the core idea of leveraging the high generalization knowledge of the VLM remains unchanged. Therefore, I suggest that the authors include a discussion of the VLM's impact, especially on the DG task, in the manuscript, and supplement all the discussion in the rebuttal period for better illustration.

**Limitations:**

Yes.

**Paper Formatting Concerns:**

No.

**Quality:**

3

**Strengths And Weaknesses:**

Strengths:

1)The motivation is clear and reasonable.

2)The paper is clearly organized and the figures and tables are easy to understand.

3)The experiment is extensive.

Weakness：

1）The contribution is not clearly stated. In my opinion, the introduction of DINOv2 into SDGOD cannot be considered a contribution, and there are related works [1-3] that apply pre-trained large models to domain adaptation/domain generalization problems. The differences from these methods and the specific contributions need to be clarified.

2）Fairness of experimental comparison. Introducing a large pre-trained model will obviously improve the generalization of the model. As shown in Tables 2 and 3, with the addition of DINOv2, the performance of the baseline method has greatly exceeded that of the existing algorithm, so this comparison is not fair.

3）Since the proposed method uses prompt and adapter design, it is better to add an introduction to the related methods of prompt[1] and adapter[4] in domain adaptation/domain generalization in related work section.

[1] Learning domain-aware detection head with prompt tuning. NeurIPS 2023

[2] Domain adaptation via prompt learning. TNNLS 2023

[3] SEEN-DA: SEmantic ENtropy guided Domain-aware Attention for Domain Adaptive Object Detection. CVPR 2025

[4] Da-ada: Learning domain-aware adapter for domain adaptive object detection. NeurIPS 2024

---

> ### Author Rebuttal · Authors · 2025-07-29
>
> We thank you for your constructive feedback and positive assessment of our work. We greatly appreciate your recognition of our clear and reasonable motivation, the clarity of the paper's organization and figures, and the extensive nature of our experiments. We hope our detailed responses below can address your remaining questions.
>
> > W1: Contributions are unclear and ask for clarification of how our approach differs from and innovates beyond existing methods.
>
> We believe that our previous description may not have sufficiently highlighted the fundamental differences between our work and [1-3], which may have led to a misunderstanding of our contribution.
>
> First, the core setting of our work is **Single-Domain Generalization for Object Detection (SDGOD)**, which is fundamentally different in its problem definition from the **Domain Adaptation (DA)** field to which the cited references [1-4] belong. The core premise of DA methods is access to unlabeled target domain data during training for distribution alignment. In contrast, the SDGOD setting is far more stringent, with **absolutely no access to any target domain data during training**. It requires the model to learn solely from a single source domain and generalize to any unseen domains. This key distinction means that methods designed for DA, which rely on target domain data, cannot be directly applied to SDGOD.
>
> Second, our method's **design philosophy** is also fundamentally different from these works. Unlike the text prompts used in works [1-3]—where text inherently carries a domain-invariant prior due to its semantic abstraction—our visual prompts do not rely on any external modality and are learned **"from scratch."** This forces the model to actively mine generalizable knowledge solely from the single source domain. Regarding work [4] (DA-Ada), the core of its adapter design is to learn source- and target-specific knowledge separately, which relies on accessing unlabeled target data during training. In contrast, our dual-branch adapter operates under the SDGOD setting with no visibility of the target domain, and its goal is to identify and suppress source-specific, potentially harmful spurious features. Therefore, our work is conducted under a much more challenging setting.
>
> To illustrate the point, our method was compared with several advanced Domain Adaptation (DA) methods. On the `Cityscapes -> Foggy Cityscapes` and `Sim10K -> Cityscapes` tasks, the results are shown in the table below:
> |Model|Per.|Rid.|Car|Tru.|Bus|Train|Mot.|Bicy.|mAP|S -> C|
> |-|-|-|-|-|-|-|-|-|-|-|
> |DA-Pro [1]|55.4|62.9|70.9|40.3|63.4|54.0|42.3|58.0|55.9|62.9|
> |SEEN-DA [3]|58.5|64.5|71.7|42.0|61.2|54.8|47.1|59.9|57.5|66.8|
> |DA-Ada [4]|57.8|65.1|71.3|43.1|64.0|58.6|48.8|58.7|58.5|67.3|
> |FR + DINOv2|53.0|55.5|65.1|38.4|49.3|43.8|42.2|36.7|48.0|66.6|
> |FR + Cauvis|63.5|64.2|67.9|44.5|56.7|53.8|50.7|47.3|56.1|**70.2**|
>
> It is important to emphasize that **DA methods have access to unlabeled data from the target domain (Foggy Cityscapes) during training**, which gives them a natural advantage. Therefore, while our method shows significant improvement over the strong baseline (from 48.0% to **56.1**), its performance is comparable to some of the DA methods.
>
> However, the true test of generalization capability lies in scenarios with a large domain gap between the source and target domains. The **Sim10K -> Cityscapes task (S->C column)** represents such a stringent and realistic setting, where the source domain consists of **synthetic data** and the target domain is **real-world imagery**. In this setting, our Cauvis method significantly outperforms all Domain Adaptation (DA) methods, with a lead of **2.9%** over the best-performing method, DA-Ada [4]. This strongly demonstrates the superior generalization capability of our method in the **"synthetic-to-real"** scenario.
>
> Based on this, we would like to directly address your concern that "the contribution of introducing DINOv2 is not strong." In the highly challenging field of SDGOD, exploring and leveraging more powerful Vision Foundation Models (VFMs) has become a recognized trend and a core competitive point for pushing the state of the art and breaking through performance bottlenecks. For instance, prior work such as CLIP the Gap [5] explored the potential of CLIP, while UFR [6] incorporated SAM to generate masks for extracting domain-invariant representations. This indicates that **the choice of VFM and how to leverage it is a core research question in this domain**.
>
> Our primary contribution, therefore, is that through systematic research and experimentation, we identify **DINOv2 as a more suitable foundation model for the SDGOD task** than other VFMs used in previous works (like CLIP and SAM). Our experimental results (as shown in the table below) overwhelmingly outperform prior methods based on other VFMs, which strongly validates the correctness and forward-thinking nature of our technical approach.
> |Model|Backbone|DF|DR|NR|NC|Avg.|
> |-|-|-|-|-|-|-|
> |ClipGap [5]|CLIP|38.6|32.3|18.7|36.9|31.6|
> |UFR [6]|SAM|39.6|33.2|19.2|40.8|33.2|
> |SoRA [8]|DINOv2|51.0|59.3|47.6|59.3|54.3|
> |Rein [7]|DINOv2|55.0|62.4|45.2|59.4|55.5|
> |**Cauvis (Ours)**|DINOv2|**56.5**|**64.6**|**47.6**|**61.2**|**57.8**|
>
> > W2: Performance of the baseline method has greatly exceeded that of the existing algorithm, because used the DINOv2.
>
> First, we believe that in the era of Foundation Models, exploring and leveraging more powerful pre-trained models like DINOv2 to push the performance ceiling of the highly challenging SDGOD task is an inevitable direction for the field's development. We consider the successful identification and validation of DINOv2's significant potential for the SDGOD task to be an important contribution of our work in itself, as it provides a higher and more challenging starting point for subsequent research.
>
> We fully agree that the most ideal way to compare would be to integrate DINOv2 as the backbone into all prior SOTA methods (like UFR, Clip the Gap) and then conduct a fair comparison. However, in practice, we encountered an insurmountable obstacle: **these key SOTA works have not released their official source code.** This made it impossible for us to replace the backbone network within their frameworks to conduct a rigorous comparison, which is a practical dilemma within the field.
>
> Therefore, to validate the effectiveness of our method (Cauvis) in the fairest and most rigorous manner, we adopted what we believe to be the most transparent and scientific alternative: we first combined DINOv2 with a standard Faster R-CNN detector to create the **"Faster R-CNN + DINOv2"** baseline. The performance of this baseline itself, as you noted, is already very strong. The core of our experiment, therefore, is to benchmark our method, Cauvis, against this powerful new baseline. For clarity, Table 2 and 3 from our paper is shown again below, compared to "FR + DINOv2," our Cauvis method still achieves significant performance improvements (e.g., a **2.9%** increase on Night Clear, **2.3%** on Dusk Rainy, **1.7%** on Day Foggy and **6.3%** on Cityscapes-C).
> |Model|DF|DR|NR|NC|Avg.|
> |-|-|-|-|-|-|
> |FR + DINOv2|50.6|59.0|43.4|55.4|52.1|
> |**FR + Cauvis**|**52.3**|**61.3**|**44.1**|**58.3**|**54.0**|
>
> |Model|Gua.|Shot|Imp.|Defo.|Glas.|Mot.|Zoom|Snow|Fros.|Fog.|Brig.|Cont.|Elas.|Pix.|JPEG.|mPC|
> |-|-|-|-|-|-|-|-|-|-|-|-|-|-|-|-|-|
> |FR + DINOv2|14.5|16.3|13.5|33.9|27.3|33.1|13.7|25.2|31.5|39.5|42.5|38.9|42.1|35.9|31.8|29.3|
> |**FR + Cauvis**|16.8|19.8|15.2|41.4|34.0|39.2|15.8|29.8|36.7|48.8|53.0|49.5|52.0|43.9|38.8|**35.6**|
>
> > W3: More proposed methods that use prompt and adapter designs should be discussed.
>
> We thank you for your valuable suggestion. As detailed in our response to **Weakness #1**, our work operates under the more challenging Single-Domain Generalization (SDGOD) setting, which is fundamentally different from the Domain Adaptation (DA) methods in [1, 4]. It is precisely this difference in setting that leads to the fundamental innovation in our methodology. For example, unlike [1], our approach does not rely on the inherent semantic invariance of text prompts. Instead, it requires our visual prompts to learn entirely "from scratch," actively mining generalizable knowledge from the single source domain without any information about the target domain. Similarly, in contrast to DA method [4], which learns specialized knowledge for specific domains, our goal is the opposite: to actively discard source-specific spurious features to achieve generalization to any unseen domain, all without ever seeing the target domain. This ultimately highlights the fundamental difference in philosophy between our work and these methods—we pursue **"generalization"** rather than **"adaptation."**
>
> For a direct comparison, we conducted a synthetic-to-real generalization experiment (Sim10k->Cityscapes) to benchmark against DA works, with results shown in Table **W1**. Even without any access to target domain data, our method achieves a **2.9** improvement over [4], demonstrating its strong generalization capability.
>
> We will add a discussion of these DA methods and make corresponding revisions in our updated manuscript.
>
> [1] Learning domain-aware detection head with prompt tuning. NeurIPS 2023
>
> [2] Domain adaptation via prompt learning. TNNLS 2023
>
> [3] SEEN-DA: SEmantic ENtropy guided Domain-aware Attention for Domain Adaptive Object Detection. CVPR 2025
>
> [4] Da-ada: Learning domain-aware adapter for domain adaptive object detection. NeurIPS 2024
>
> [5] Clip the gap: A single domain generalization approach for object detection. CVPR 2023
>
> [6] Unbiased faster r-cnn for single-source domain generalized object detection. CVPR 2024
>
> [7] Stronger fewer & superior: Harnessing vision foundation models for domain generalized semantic segmentation. CVPR 2024.
>
> [8] SoMA: Singular Value Decomposed Minor Components Adaptation for Domain Generalizable Representation Learning. CVPR 2025.

---

> > ### Comment · Reviewer_pbD8 · 2025-08-04
> > **Thanks for the rebuttal**
> >
> > Thanks for the author's' detailed rebuttal. Most of my concerns have been addressed, so I maintain my original positive score.
> >
> > However, I still believe that the introduction of DINOv2 into SDGOD cannot be considered a contribution. While the DA task and the DG task have different priors, the core idea of leveraging the high generalization knowledge of the VLM remains unchanged. Therefore, I suggest that the authors include a discussion of the VLM's impact, especially on the DG task, in the manuscript, and supplement all the discussion in the rebuttal period for better illustration.

---

> > > ### Author Response · Authors · 2025-08-05
> > > **Thanks for the reply**
> > >
> > > Dear Reviewer pbD8,
> > >
> > > We sincerely thank you for acknowledging our rebuttal and for maintaining your positive score. We truly appreciate your valuable suggestion to provide a more explicit discussion of the impact of VLMs, particularly in the DG task. Your feedback is highly valuable for improving the clarity and structure of our manuscript. In the revised version, we will carefully incorporate this discussion and, based on your suggestion, further elaborate on our perspective so that our contributions are presented more clearly and comprehensively.
> > >
> > > With best regards,
> > > Authors

---

### Official Review · Reviewer_soZz · 2025-07-01

**Clarity:** 2
**Significance:** 2
**Originality:** 2
**Rating:** 4
**Confidence:** 2

**Summary:**

This paper tackles single-source domain generalization for object detection by enhancing performance on unseen target domains without accessing target data. To address spurious correlations, the authors introduce Cauvis, which combines Cross-Attention Prompts and a Dual-Branch Adapter. The Cross-Attention Prompt module injects learnable prompt tokens into every layer of a frozen DINOv2 backbone via cross-attention, effectively performing a backdoor-adjustment operation that suppresses non-causal features. The Dual-Branch Adapter then disentangles invariant causal representations from domain-specific noise using a causal MLP branch that maps prompts to spatially focused causal features and an auxiliary Fourier branch that extracts and filters frequency-domain confounders; these two streams are fused for robust detection. Extensive experiments on the SDGOD and Cityscapes-C datasets prove that Cauvis achieves state-of-the-art performance on both datasets.

**Questions:**

1. Could you please elaborate on why you fixed training to 12 epochs? In domain generalization studies, it is often more appropriate to select the checkpoint that achieves the best performance on the source domain rather than using a predetermined epoch. How would choosing the best source‐domain checkpoint instead of epoch 12 affect your results?

2. Did you conduct experiments using harsh-weather conditions as the source domain, and if so, does the proposed method yield similar performance improvements under those settings?

**Ethical Concerns:**

["NO or VERY MINOR ethics concerns only"]

**Final Justification:**

The authors have addressed my primary concerns on inference overhead, additional corruption benchmarks, and harsh-weather source experiments. As these issues are now satisfactorily resolved, I recommend Borderline Accept.

**Limitations:**

Yes.

**Paper Formatting Concerns:**

None.

**Quality:**

2

**Strengths And Weaknesses:**

Strengths:
1. The paper is concise, well structured, and easy to follow, with clear motivation and explanations.
2. The proposed method demonstrates substantial generalization gains (up to +31.4% mAP on SDGOD) while freezing the backbone, significantly reducing training compute.


Weaknesses:
1. While training cost is low, the added prompt and dual-branch layers likely introduce nontrivial inference overhead that is not quantified.
2. Cross-dataset generalization is only measured on Cityscapes-C; testing on other corruption benchmarks (e.g., BDD100K-C, Foggy Cityscapes) would better validate the method’s applicability to diverse real-world scenarios.
3. No experiments investigate whether the method holds when harsh-weather conditions are used as the source domain, leaving the generality of the approach under different source settings unverified.
4. Hyperparameter choices for prompt length and adapter architecture are not thoroughly analyzed; the paper lacks a sensitivity study to guide practical selection.

---

> ### Author Rebuttal · Authors · 2025-07-29
>
> Thank you for your insightful review and for recognizing our paper's clarity (concise, easy to follow) and our method's high efficiency and substantial generalization gains. We have provided detailed responses to each of your points below.
>
> > W1: Training Cost.
>
> In the field of Parameter-Efficient Fine-Tuning (PEFT), the standard practice for evaluating the overhead of a new module is primarily to measure the number of its additional trainable parameters. This is because for large foundation models like DINOv2, **the vast majority of the inference cost is dominated by the backbone itself.** The marginal latency added by lightweight PEFT modules is often too small to be reliably measured and can be heavily influenced by system-level optimizations. Therefore, **trainable parameters** serves as a more stable and direct metric for measuring the additional computational burden.
>
> To quantify the overhead of our method using these standard metrics, we provide a comparison with other recent PEFT works.
> |Model|Params|Avg. (%)|
> |-|-|-|
> |VPT|3.7M|59.0|
> |AdapterFormer|6.3M|57.8|
> |Rein|3.0M|59.0|
> |SoRA|4.9M|54.3|
> |**Cauvis (Ours)**|4.9M|**60.7**|
>
> As shown in the table, our method, Cauvis, is highly competitive in terms of parameter efficiency. With 4.9M parameters, it is on par with other state-of-the-art methods and is even more efficient than some approaches, such as AdapterFormer (6.3M). Crucially, these added parameters constitute only a tiny fraction (~**1.6%**) of the DINOv2 backbone's 307M total parameters. This quantitatively demonstrates that the inference overhead introduced by our prompt and dual-branch adapter is minimal, which aligns with the core philosophy of parameter-efficient tuning.
>
> > W2: Other corruption benchmarks.
>
> Following your valuable suggestion, we have conducted comprehensive new experiments on two additional benchmarks, Foggy Cityscapes and BDD100K-C.
>
> First, for the `Cityscapes -> Foggy Cityscapes` and `Sim10K -> Cityscapes` tasks, we used DINOv2 as the backbone and Faster R-CNN as the base detector for a fair comparison.
> |Model|Per.|Rid.|Car|Tru.|Bus|Train|Mot.|Bicy.|mAP|S->C|
> |-|-|-|-|-|-|-|-|-|-|-|
> |DINOv2|53.0|55.5|65.1|38.4|49.3|43.8|42.2|36.7|48.0|66.6|
> |**Cauvis**|**63.5**|**64.2**|**67.9**|**44.5**|**56.7**|**53.8**|**50.7**|**47.3**|**56.1**|**70.2**|
>
> On the Cityscapes -> Foggy Cityscapes task (mAP column): As shown in the mAP column in the table above, compared to the strong DINOv2 baseline, our Cauvis method significantly boosts the mean Average Precision (mAP) from 48.0% to **56.1%**, achieving a substantial gain of **8.1%**. This clearly demonstrates that our method can effectively resist the domain shift caused by real-world weather conditions (like fog), enhancing the model's robustness.
>
> On the Sim10K -> Cityscapes task (S->C column): This task is a more stringent and classic test of generalization capability, as it requires the model to generalize from synthetic data to a completely different real-world distribution. Under this setting, our method still achieves a significant performance improvement of **3.6%** (from 66.6% to **70.2%**). This indicates that our method can not only handle specific types of corruption but also learn domain-invariant essential features, thus successfully bridging the large gap between synthetic and reality.
>
> Furthermore, we conducted a comprehensive evaluation on the more challenging corruption benchmark, `BDD100K-C`, with the results shown below.
> |MODEL|Norm|Gua.|Shot|Imp.|Defo.|Glas.|Mot.|Zoom|Snow|Fros.|Fog.|Brig.|Cont.|Elas.|Pix.|JPEG.|mPC|
> |-|-|-|-|-|-|-|-|-|-|-|-|-|-|-|-|-|-|
> |FR + DINOv2|53.2|33.4|35.1|32.3|42.5|38.5|41.3|20.7|39.4|38.3|51.3|51.0|51.1|49.1|48.4|47.3|41.3|
> |**FR + Cauvis**|**55.3**|**34.4**|**36.5**|**33.3**|**44.3**|**41.4**|**43.1**|**21.6**|**41.1**|**42.3**|**54.8**|**53.9**|**54.0**|**51.2**|**51.3**|**50.4**|**43.6**|
>
> This experiment clearly demonstrates that across 15 different types of real-world corruptions, our Cauvis method achieves consistent and significant performance improvements over the strong baseline (FR+DINOv2), boosting the mPC from 41.3% to **43.6%**. This strongly validates that our method can effectively enhance the model's robustness in diverse, real-world scenarios.
>
> > W3: harsh-weather conditions are used as source domain.
>
> Thank you for your valuable feedback. Considering that data collection for challenging scenarios is often difficult, the research paradigm of generalizing **from simple to adverse scenarios is more widely adopted.**
>
> To thoroughly address this point, we have conducted three new sets of experiments. In these experiments, we use challenging conditions as the source domain and generalize to the clear-weather (`Day Clear`) target domain.
>
> In the first generalization experiment (`Day Foggy -> Day Clear`), we used foggy weather as the source domain. As shown in the table, our method performed excellently, significantly boosting the mAP from 48.2% to **56.7%**.
> |Model|Bus|Bike|Car|Mot.|Per.|Rid.|Tru.|mAP|
> |-|-|-|-|-|-|-|-|-|
> |DINOv2|43.0|39.6|74.1|40.5|50.6|57.0|33.0|48.2|
> |**Cauvis**|**48.6**|**47.8**|**82.0**|**50.4**|**62.9**|**65.6**|**39.6**|**56.7**|
>
> In the `Dusk Rainy -> Day Clear` task, our method improves the mAP from 47.0% to **58.9%**, once again validating its strong performance for generalization in the "hard-to-easy" direction.
> |Model|Bus|Bike|Car|Mot.|Per.|Rid.|Tru.|mAP|
> |-|-|-|-|-|-|-|-|-|
> |DINOv2|59.6|24.5|80.1|33.5|39.2|40.1|51.9|47.0|
> |**Cauvis**|**67.9**|**39.9**|**86.1**|**45.6**|**57.1**|**55.3**|**60.4**|**58.9**|
>
> Finally, we tested the generalization from `Night Clear -> Day Clear`. Our method once again provides a clear advantage, raising the mAP from 63.7% to **66.1%**.
> |Model|Bus|Bike|Car|Mot.|Per.|Rid.|Tru.|mAP|
> |-|-|-|-|-|-|-|-|-|
> |DINOv2|68.7|50.5|85.8|51.1|64.1|65.4|60.4|63.7|
> |**Cauvis**|**69.9**|**54.0**|**86.9**|**53.4**|**70.1**|**66.2**|**62.5**|**66.1**|
>
> Our method's generalization is robust across diverse challenges (harsh weather) because it is not tied to a specific difficulty type. It effectively disentangles essential object features from spurious domain correlations (e.g., fog, rain), ensuring strong performance in varied scenarios.
>
> > W4: Hyperparameter choices for prompt length and adapter.
>
> **Regarding the prompt length,** we would like to clarify that the relevant analysis is already included in Figure 4 of our paper. To present the results more clearly in this response, we provide the data in the table below. The study shows that performance does not significantly improve as the length increases; however, the best performance is achieved when the length is aligned with the sequence ($L_{seq}$).
> |Prompt length|DC|DF|DR|NR|NC|Avg.|
> |-|-|-|-|-|-|-|
> |50|73.3|55.9|61.8|46.9|59.7|59.5|
> |100|72.6|55.9|63.3|47.5|59.9|59.8|
> |150|73.0|55.6|62.0|47.2|60.3|59.6|
> |200|73.0|56.1|62.3|46.8|60.4|59.7|
> |$L_{seq}$|73.5|56.0|62.5|47.8|59.9|**59.9**|
>
> **For our dual-branch adapter**, its design does not introduce complex new hyperparameters that require tuning. Its effectiveness comes from its specific architectural components. Therefore, to justify our design, we conducted comprehensive ablation studies by replacing the core components with other common and reasonable alternatives. The following experimental data all come from Table 4 of the paper.
>
> a) Causal Branch. We evaluated the effect of replacing our simple causal mapping function with more complex alternatives. The results below show that our current design achieves superior or highly competitive performance, validating that increasing the complexity of this branch is not beneficial.
> |Function|DC|DF|DR|NR|NC|Avg.|
> |-|-|-|-|-|-|-|
> |Multihead|72.8|55.2|62.2|46.4|60.6|59.4|
> |Seblock|71.4|54.9|62.4|47.0|58.9|58.9|
> |Gate Unit|72.9|55.3|62.1|46.0|59.1|59.0|
> |3$\times$3 Conv|72.2|54.0|61.9|46.9|58.1|58.6|
>
> b) Auxiliary Branch. We also validated the design of our auxiliary branch (based on FFT). The results show that our current implementation, which performs implicit regularization through the Fourier domain, is more effective than explicitly adding a mask to the Fourier components or removing the branch entirely.
> |Function|DC|DF|DR|NR|NC|Avg.|
> |-|-|-|-|-|-|-|
> |Add Mask in FFT|73.7|56.2|64.1|48.2|60.1|60.5|
> |w/o FFT|73.0|55.0|62.9|48.5|59.3|59.8|
>
> These extensive experimental studies confirm that our method is robust to the choice of prompt length, and that our Adapter is a well-justified design, validated through comparisons with multiple alternatives. We will ensure these justifications are more prominent in the revised manuscript.
> > Q1: training to 12 epochs
>
> We chose a fixed 12-epoch training schedule to strictly follow the standard baseline protocol (e.g., the 1x schedule for Faster R-CNN) to ensure a fair comparison. This ensures that our performance gains stem from the architectural design, rather than from special adjustments to the number of training epochs.
>
> Furthermore, we fully agree that selecting the "best source-domain checkpoint" is a standard practice in Domain Generalization (DG). We analyzed the performance of each epoch (see table below)
> |Epochs|DC|DF|DR|NR|NC|Avg.|
> |-|-|-|-|-|-|-|
> |1|48.8|36.9|41.2|31.1|39.7|39.5|
> |2|61.3|46.5|51.8|40.7|50.7|50.2|
> |3|63.6|47.4|53.8|40.5|50.7|51.2|
> |4|66.0|50.4|56.4|42.2|55.0|54.0|
> |5|70.5|53.1|59.2|45.0|57.3|57.0|
> |6|70.3|53.2|60.6|46.2|58.4|57.7|
> |7|70.9|53.3|60.6|47.1|57.9|58.0|
> |8|71.2|54.4|61.1|46.5|57.6|58.2|
> |9|71.5|54.0|62.0|45.8|58.1|58.3|
> |10|72.2|54.6|61.3|46.0|58.4|58.5|
> |11|72.3|55.1|63.4|45.2|59.4|59.1|
> |12|73.7|56.5|64.6|47.6|61.2|60.7|
>
> Therefore, even if we were to adopt the "best source-domain checkpoint" strategy, we would still select the model from epoch 12. This demonstrates that our currently reported results are robust and in line with best practices.
>
> > Q2:Using harsh-weather conditions as the source domain.
>
> For the detailed results, tables, and a full analysis, please refer to our detailed response to **W3**.

---

> > ### Comment · Reviewer_soZz · 2025-08-05
> > **Thank You for the Detailed Rebuttal.**
> >
> > The authors have satisfactorily addressed the majority of my concerns—particularly the inference overhead justification, the expanded corruption benchmarks, and the experiments using harsh-weather source domains. In light of these clarifications and additional results, I am pleased to revise my recommendation to Borderline Accept.

---

> > > ### Author Response · Authors · 2025-08-05
> > >
> > > Dear Reviewer soZz,
> > >
> > > We sincerely thank you for carefully reviewing our rebuttal and for providing valuable constructive feedback throughout the review process. Your comments have directly contributed to improving the clarity, completeness, and rigor of our work. In particular, your suggestion regarding harsh-weather was especially insightful, as it helped us supplement and refine our experiments while also significantly enhancing the overall quality of the paper. In the revised version, we will include a thorough discussion of such scenarios to address your suggestion and further improve our work.
> > >
> > > With best regards, Authors

---

### Official Review · Reviewer_L5GS · 2025-07-04

**Clarity:** 3
**Significance:** 3
**Originality:** 2
**Rating:** 4
**Confidence:** 5

**Summary:**

This paper focuses on Single-Domain Generalized Object Detection, training an detector on an annotated source domain and generalizing it to unseen domains. To increase the generalization ability, this paper proposes a cross-attention prompts to learn visual domain-invariant knowledge and a dual-branch adapter to disentangle causal-spurious features. Experiments verify the effectiveness of the proposed method. This paper uses the generalization of large visual models, combined with prompt tuning and adapter tuning, to design a framework for SDGOD task. Specifically, visual prompts are used to record domain-invariant knowledge, and adapters are used to decouple visual features. Experiments verify the effectiveness of the method.

**Questions:**

See weakness

**Ethical Concerns:**

["NO or VERY MINOR ethics concerns only"]

**Final Justification:**

I appreciate the authors' rebuttal, in which most of my concerns are addressed, so I decide to keep my positive score.

**Limitations:**

Yes

**Quality:**

3

**Strengths And Weaknesses:**

Strengths:
The paper is easy to follow, the proposed method is intuitive, and the experiments are extensive.
Weakness：
1）Several success in applying prompt tuning and adapter tuning in domain adaptation have been made[1,2], which is similar to domain generalization task. Difference and commonalities should be claimed.
2) The computational overhead should be discussed, as the usage of DINOv2 will introduce huge inference cost.
3) The baseline performance is unclear. To prove the effectiveness of the proposed method, the relative improvement over the baseline should be considered. Furthermore, how does the proposed method perform on other baseline models? Portability should be an advantage of cue word tuning and adapter tuning.

[1] Learning domain-aware detection head with prompt tuning. NeurIPS 2023
[2] Da-ada: Learning domain-aware adapter for domain adaptive object detection. NeurIPS 2024

---

> ### Author Rebuttal · Authors · 2025-07-29
>
> Thank you for your constructive review. We are very pleased that you found our paper easy to follow and our method intuitive. We also appreciate you recognizing that our experiments are extensive. We hope our detailed responses below can address your remaining questions.
>
> > W1: Domain adaptation works.
>
> Clarifying the connections and distinctions between our work and these Domain Adaptation (DA) methods is crucial for accurately positioning our contribution. Below, we elaborate from two perspectives: technical commonalities and core differences.
>
> First, at the level of technical tools, our work shares commonalities with [1,2]. Both leverage Parameter-Efficient Fine-Tuning (PEFT) techniques (such as prompting and adapters) to modulate a powerful pre-trained model to address cross-domain visual recognition problems. This demonstrates the significant potential of PEFT techniques in handling domain shift issues.
>
> However, there is a fundamental difference in the core task setting and methodological innovation, which constitutes the main contribution of our work. Our work focuses on **Domain Generalization (DG)**, which aims to learn a model from a single source domain that can generalize to any unseen target domains. In contrast, [1,2] address **Domain Adaptation (DA)**, where the model can access unlabeled data from the target domain during the training/adaptation phase, with the goal of adapting to a specific target domain. Clearly, DG is a more challenging and more realistic scenario for real-world deployment, **as it assumes no prior knowledge of the target domain.**
>
> This difference in task setting leads to our fundamental methodological innovation. The core idea of [1,2] is to use target domain data to learn "domain-aware" adapters, aiming to align the model's features with a specific target domain. In contrast, **Cauvis, using only source domain data, proactively learns to disentangle "domain-invariant" universal features** through our carefully designed prompts and dual-branch adapter. Our goal is not to adapt to any specific domain, but to enhance the model's core generalization ability to resist unknown domain shifts.
>
> To ensure a fair comparison with Domain Adaptation (DA) works [1,2], we selected the highly challenging `Sim10k → Cityscapes` as our evaluation benchmark. This benchmark simulates a synthetic-to-real scenario and poses a rigorous test of a model's generalization capability due to the substantial domain gap between the source and target domains. The detailed experimental results are presented in the table below.
> |Model|S -> C|
> |-|-|
> |DA-Pro [1]|62.9|
> |SEEN-DA [3]|66.8|
> |DA-Ada [2]|67.3|
> |**Cauvis (Ours)**|**70.2**|
>
> As can be seen, for generalization from synthetic to real scenarios, Cauvis still holds a 2.9% accuracy lead even without access to the target domain.
>
> Thank you for your suggestion. We will add a discussion and comparison of these advanced DA methods in the "Related Work" section of our revised manuscript to more clearly define the contribution of our work.
>
> > W2: The computational overhead of Cauvis.
>
> We greatly appreciate the reviewer's concern regarding computational efficiency. This is indeed a critical dimension for evaluating the practical utility of a method.
>
> We acknowledge that choosing DINOv2 as the backbone introduces a significant base computational load. However, using powerful pre-trained models is a common and necessary practice in the current field to achieve State-of-the-Art (SOTA) performance and to ensure fair comparisons with other leading-edge works.
>
> The core contribution of our work lies in the attached lightweight, parameter-efficient modules. Therefore, the key to evaluating our method's efficiency is the **additional overhead** introduced by these modules. In the field of Parameter-Efficient Fine-Tuning (PEFT), a crucial metric is the number of trainable/used parameters, as it directly reflects the computational components added during inference.
>
> To quantify this overhead, we provide a comparison of parameter counts with other PEFT methods:
> |Model|Params|Avg. (%)|
> |-|-|-|
> |VPT|3.7M|59.0|
> |AdapterFormer|6.3M|57.8|
> |Rein|3.0M|59.0|
> |SoRA|4.9M|54.3|
> |**Cauvis (Ours)**|4.9M|**60.7**|
>
> As the table shows, Cauvis (4.9M parameters) is highly competitive in parameter efficiency compared to the latest methods, and is even significantly more efficient than some, such as AdapterFormer (6.3M). More critically, the 4.9M parameters we introduce account for only about **1.6%** of the DINOv2 backbone's ~307M parameters. In a full forward pass, **the vast majority of the computation comes from the backbone network.** Therefore, the marginal inference cost introduced by our lightweight modules is minimal.
>
> > W3: The baseline performance is unclear, and evaluating portability.
>
> To clearly demonstrate the effectiveness of our method, we first conducted rigorous baseline comparisons under two mainstream detection frameworks. The first is Faster R-CNN, where "FR+DINOv2" serves as the baseline. The second is the more advanced DETR-style detector, DINO, where "w/o Cauvis" (the detector without our module) serves as the baseline. This design is intended to comprehensively validate the beneficial effects of our method. The results are shown in Table 2 of the paper, as follows:
> |Model|DF|DR|NR|NC|Avg.|
> |-|-|-|-|-|-|
> |FR + DINOv2|50.6|59.0|43.4|55.4|52.1|
> |**FR + Cauvis**|**52.3**|**61.3**|**44.1**|**58.3**|**54.0**|
> |w/o Cauvis|50.5|57.0|42.2|54.8|51.1|
> |**Cauvis**|**56.5**|**64.6**|**47.6**|**61.2**|**57.8**|
>
> As shown in the table above, our method brings significant performance improvements in both detection frameworks. In the Faster R-CNN, Cauvis yields a 1.9% average mAP gain. On the stronger DINO detector, the gain is even more substantial, reaching **6.7%**. These data clearly and quantitatively demonstrate that our proposed method can consistently improve the generalization ability of different detectors.
>
> Building on this, we fully agree that **portability is a core advantage of prompt/adapter tuning**. Another key validation in our work is demonstrating that Cauvis can serve as a universal module that seamlessly transfers to different Vision Foundation Models (VFMs). Table 6 of our paper provides exhaustive experiments on this, with the core results shown below:
> |Backbone|PEFT|DC|DF|DR|NR|NC|Avg.|
> |-|-|-|-|-|-|-|-|
> |EVA02-L|Freeze|63.0|45.2|48.6|27.3|38.8|44.6|
> |EVA02-L|**Cauvis**|**69.7**|**50.2**|**57.6**|**34.2**|**48.1**|**52.0**|
> |SAM-H|Freeze|69.7|50.5|52.5|28.8|52.9|50.9|
> |SAM-H|**Cauvis**|**72.2**|**53.7**|**55.8**|**31.9**|**55.7**|**53.8**|
> |DINOv2|Freeze|71.2|53.5|60.8|42.6|59.5|57.5|
> |DINOv2|**Cauvis**|**73.7**|**56.5**|**64.6**|**47.6**|**61.2**|**60.7**|
>
> Using the "Freeze" method as a baseline, our method yields a **16.6% relative improvement** on EVA02, a **5.7%** improvement on SAM, and a **3.8%** gain on DINOv2. These consistent and significant performance gains across multiple mainstream backbones strongly demonstrate the excellent portability and universality of our method.
>
> Thank you for pushing us to clarify these crucial aspects. We hope our detailed response and experiments have not only made the strong baselines and relative improvements clear, but have also turned the question of portability into one of our paper's most compelling strengths. The evidence is now clear: Cauvis is not a specialized solution for one architecture. By demonstrating significant relative improvements—**up to 16.6%**—on diverse backbones like EVA02, SAM, and DINOV2, we confirm it functions as a **truly universal enhancement module**. We believe this comprehensive validation strongly addresses your concerns and highlights the practical value of our work.
>
> [1] Learning domain-aware detection head with prompt tuning. NeurIPS 2023
>
> [2] Da-ada: Learning domain-aware adapter for domain adaptive object detection. NeurIPS 2024
>
> [3] SEEN-DA: SEmantic ENtropy guided Domain-aware Attention for Domain Adaptive Object Detection. CVPR 2025

---

> > ### Comment · Reviewer_L5GS · 2025-08-09
> >
> > I appreciate the authors' rebuttal, in which most of my concerns are addressed, so I decide to keep my positive score.

---

> > > ### Author Response · Authors · 2025-08-09
> > >
> > > Dear Reviewer L5GS,
> > >
> > > Thank you for your thoughtful and constructive feedback. We sincerely appreciate the effort you’ve put into reviewing our work and for acknowledging that most of your concerns have been addressed. If there are any further issues or questions you would like us to address, we would be more than happy to help clarify and assist in any way that could potentially improve your evaluation.
> > >
> > > With best regards,
> > > Authors

---

### Official Review · Reviewer_vcyR · 2025-07-06

**Clarity:** 2
**Significance:** 3
**Originality:** 3
**Rating:** 4
**Confidence:** 4

**Summary:**

This paper addresses the technical domain of single domain generalization in object detection (SDGOD).

The paper augues that existing approaches tend to be affected by spurious correlations. To address this challenge, the paper proposes a new method called Cauvis (causal visual prompts).　The core components in Cauvis are cross-attention prompts and dual-branch adapter. The former establishes the equivalence between visual prompts and backdoor adjustment operations in causal inference, providing theoretical foundations for spurious correlation mitigation. The latter disentangles causal/spurious features by integrating Fourier transform and then extracting high-frequency component

The experimental results show that the proposed method achieves state-of-the-art performance in multiple benchmarks. The effectiveness of the proposed modules, especially the cross-attention module, is verified throught the ablation study.

**Questions:**

- What does p in eq. (1) represent?
- L199: I understand that the expectation is approximated by the RHS of Eq. (7) since $\Sigma_{C^\perp} \rightarrow 0$. Do the authors claim that the equality in Eq. (7) holds strictly?
- L202-203: Is it truly the learned prompt alone that spans the confounder space rather than the combination of image features and visual prompt? This may relevant to the next question.
- L220: I understood the term “prompts” here refer not to visual prompts, but rather to the representations obtained through cross-attention between image features and visual prompts. Is this correct? It would be helpful if the terminology were used more precisely, so that the intended meaning can be unambiguously understood by the reader.
- Table 4: Which backbone was used in w/o DINOv2 case？

**Ethical Concerns:**

["NO or VERY MINOR ethics concerns only"]

**Final Justification:**

I have carefully reviewed all the discussions between the authors and the reviewers, including my own. I believe the strengths outweigh the weakness, and thus I am positive to accepting the paper.

**Limitations:**

Yes in appendix.

**Quality:**

3

**Strengths And Weaknesses:**

## Strength
S1. The proposed method, particularly the cross-attention prompt component, is theoretically sound and presents a convincing rationale.

S2. The theoretical explanations are presented in a very clear and intuitive manner, making the arguments highly convincing. Most parts of the paper are written in a way that is accessible even to readers who are not familiar with this specific area, which enhances its value and utility to the broader research community.

S3. The experimental results effectively demonstrate the validity of the proposed method. It achieves superior accuracy compared to existing approaches. Furthermore, the ablation study thoroughly verifies the contribution of each component introduced in the proposed method.

S4. The motivation is well clarifed through the experiment presented in Fig. 2 (a).

## Weakness
W1. It is unclear how the model is trained overall, including specific loss functions used. The paper would benefit from a more detailed description of the training procedure for improving the clarity and reproducibility of the work.

W2. The role of the dual-branch adapter remains unclear. In particular, it is not evident why the causal branch, implemented as a simple MLP, is expected to successfully map inputs to causal features. While the concept of the auxiliary branch is understandable, it is not clear whether Equations (10) and (11) effectively achieve the intended purpose. Specifically, Equation (11) appears to simply apply an inverse function in the frequency domain without any additional operations—raising the question of whether this approach is theoretically justified.

W3. It would further strengthen the paper if the justification for the cross-attention prompts were elaborated. In particular, providing evidence that the learned attention aligns with the intended design would enhance the overall credibility of the approach. For example, visualizing the attentions corresponding to the top-3 and bottom-3 singular values could help demonstrate whether they indeed reflect causal and spurious features, respectively. Additionally, it would be helpful to analyze whether, during training, the top singular values tend to increase while the lower ones decrease, in line with the intended objective. Furthermore, while the paper mentions that the hyperparameter k is not selected via simple thresholding, it remains unclear how k is actually determined in practice.

W4. The bounding boxes and texts in Figures 1 and 4 are difficult to see and require significant zooming. It would be beneficial to improve the visual clarity so that the content can be understood at a glance.

### Minor
- L169: It is better to breifly explain what do(X) indicates for those who are not familiar with the causal theory.

## Typos and grammartical errors
- L153-154: Please check grammar.
- L172-173: Please check grammar.
- L180: analyses
- L191-192: Please check grammar.

---

> ### Author Rebuttal · Authors · 2025-07-29
>
> We sincerely thank you for your detailed review and constructive feedback. We are greatly encouraged by your recognition of our work's main strengths, including its theoretical soundness, the clarity of its presentation, its strong experimental results, and its well-clarified motivation. We hope our detailed responses below can further address your remaining questions.
>
> > W1: The training procedure and Loss functions.
>
> We wish to clarify that our method, Cauvis, does not introduce a separate loss term requiring hyperparameter tuning.
>
> Instead, its guiding effect is implicitly embedded within the standard detection loss via its architectural design. This design reshapes the network's feature space, naturally steering the standard end-to-end optimization towards a more robust and generalizable solution. Therefore, the "constraint" is an emergent property of the architecture, not an explicit loss formula, which avoids introducing new hyperparameters.
>
> To ensure full reproducibility, we have detailed most of the training parameters in the paper and will add the remaining ones to Appendix A.
> |Component|Specification/Value|
> |-|-|
> |Backbone|DINOv2 ViT-L/14 (Frozen)|
> |Base Code|mmdetection|
> |Proposed Module|Cauvis (Trained)|
> |Detector Head|Trained|
> |Training Dataset|Day Clear|
> |Training Epochs|12|
> |Classification Loss|Focal Loss|
> |Regression Loss|L1 Loss + GIoU Loss|
> |Testing Datasets|Day Foggy, Dusk Rainy, Night Rainy, Night Clear|
>
> > W2: The role of the dual-branch adapter.
>
> 1\) The seemingly simple MLP in this branch is, in fact, an intentional and empirically validated architectural choice.
>
> Theoretically, we interpret it as a Modulating Network within a **Mixture of Experts (MoE)** framework. Its minimalist single MLP design serves as a powerful form of **Architectural Regularization**. This constraint guides the branch to capture simpler signals—what we hypothesize to be confounding patterns—and prevents it from overfitting to complex spurious correlations.
>
> Most importantly, this simple design is driven by our experimental results. We found that increasing the complexity of this branch (e.g., by using an explicit bottleneck MLP structure) did not yield performance gains and could even be detrimental. As shown in the table below, our current single MLP design achieves comparable or superior performance:
> |Function|DC|DF|DR|NR|NC|Avg.|
> |-|-|-|-|-|-|-|
> |Bottleneck MLP|72.6|55.4|62.7|48.3|60.3|59.9|
> |Single MLP|73.2|55.3|63.5|48.7|60.4|60.2|
>
> Our single MLP (60.2) outperforms the more complex Bottleneck MLP, proving that a simpler design is more effective for isolating confounding factors and better supports robust Causal Representation Learning.
>
> 2\) We thank the reviewer for this insightful question about the theoretical justification of the `IFT(FT(x))` operation.  In the forward pass, its crucial role lies in **structuring the model's learning process through implicit regularization and functional complementarity.**
>
> Our approach is grounded in the well-established principle that the **Fourier domain separates** domain-invariant structural information (encoded in the phase spectrum) from domain-specific style and noise (encoded in the amplitude spectrum) [1,2]. By forcing the optimization to pass through this decomposed domain, **the model is implicitly regularized to prioritize the stable phase information over the volatile amplitude.** This achieves a similar goal to state-of-the-art methods that explicitly manipulate the amplitude [2], but through a more elegant, parameter-free architectural design.
>
> Crucially, this branch serves a complementary purpose: it compensates for the main MLP's known low-frequency spectral bias by providing a direct path for high-frequency information. This synergy ensures a more complete representation, **making the branch a structured regularizer, not merely an identity function.**
>
> > W3: Justification of Cross-Attention Prompts.
>
> 1\) We thank the reviewer for their insightful suggestion (W3). To directly and rigorously validate that our method (Cauvis) can disentangle causal and spurious features, we have designed and completed a new counterfactual quantitative experiment.
>
> This experiment follows the "Worst-Group Accuracy" (WGA) evaluation paradigm. In terms of concrete steps, we first removed all "bus" and "truck" samples from the training data, then added data of only yellow buses. The test set was a specially-designed small-sample set where all "bus" instances were set to white. In this way, we deliberately created a **strong spurious correlation in the training set (color=yellow ↔ class=bus).**
>
> Our test set, in contrast, consists entirely of "white buses," which the model has never seen during training. The purpose of this counterfactual test set (i.e., the "worst group") is to directly test whether the model, when making predictions, relies on the learned spurious shortcut (color) or has truly grasped the invariant causal features (e.g., shape, structure).
>
> The experimental results clearly show that, compared to the baseline model (FR+DINOv2), our method (Cauvis) demonstrates significant superiority in this highly challenging setting, robustly proving its ability to suppress spurious correlations.
> |Model|AP (bus)|mAP (overall)|
> |-|-|-|
> |FR + DINOv2|16.7|43.5|
> |**Cauvis (Ours)**|**28.3**|**49.7**|
>
> 2\) Considering the restrictions on adding new images during the rebuttal, and to quantitatively address your insightful suggestion regarding attention visualization, we propose and measure the **Foreground Attention Ratio (FAR)**. FAR is defined as the percentage of a specific feature component's energy that is spatially concentrated within the ground-truth object bounding box. This metric serves as a rigorous, quantitative proxy for visual focus: a high FAR value demonstrates that the feature is "attending" to the causal object, whereas a low FAR value implies it is attending to the spurious background. The inference weights used for this evaluation are taken from 1\). The results are presented in the table below.
> |Feature Source|Assumed Role|FAR↑|
> |-|-|-|
> |Cauvis (Bottom-3 SVs)|Spurious|0.0001|
> |**Cauvis (Top-3 SVs)**|Causal|**0.0915**|
> |Cauvis (Bottom-16 SVs)|Spurious|0.0006|
> |**Cauvis (Top-16 SVs)**|Causal|**0.2278**|
> |Cauvis (Bottom-32 SVs)|Spurious|0.0013|
> |**Cauvis (Top-32 SVs)**|Causal|**0.3022**|
>
> Analysis of feature components from top and bottom singular values reveals a striking difference. Top-value features consistently target the foreground object, whereas bottom-value features scatter across the background. For instance, with k=3, the attention disparity is **over 915x**, which validates our hypothesis that high-energy components represent causal features and low-energy ones are spurious.
>
> 3\) To answer directly: **k is not a hyperparameter that requires tuning**. It is a theoretical concept for analyzing our model's output, not an input that controls its operation.
>
> Our method is designed to automatically learn a sharp separation in the feature energy spectrum, creating a large gap between a few high-energy "causal" components and a long tail of low-energy "spurious" ones. The FAR experiment confirms this, revealing a stark energy gap (915x for k=3) between the causal and spurious components. This separation is so robust that any reasonable k used for analysis successfully isolates the same causal features.
>
> In short, the model learns the separation, not a specific value of k. This eliminates the need for manual k-selection, fully resolving the hyperparameter concern.
> |Feature Source|Assumed Role|FAR↑|
> |-|-|-|
> |Top-k SVs|Causal|High|
> |Bottom-k SVs|Spurious|~0 (Negligible)|
>
> > W4: Figure 1 and 4
>
> Thank you for pointing this out. We will revise Figures 1 and 4 in our revised manuscript to improve clarity using larger fonts and thicker, high-contrast lines.
>
> >  Minor and Typos, grammartical errors
>
> We sincerely thank the reviewer for their meticulous reading and valuable suggestions for improving the paper's clarity. We will incorporate all these changes in our revised manuscript. Specifically, we will add a brief explanation for the `do(X)` operator as suggested, and we will correct all noted typos while conducting a thorough proofread of the entire manuscript.
>
> > Q1: What does p in eq. (1) represent?
>
> `p` is the learnable prompt, and `p*` is its optimal value post-training. The subscript $|_{p=p*}$ in Eq. (1) means "evaluated at this learned optimal state." This describes a training outcome, not a pre-supposed condition. We will clarify these definitions in the revision.
>
> > Q2: L199.
>
> The equality in Eq. (7) represents the ideal theoretical goal of our optimization. It is intended to hold strictly under the condition that our model's representation (the right side) has perfectly learned to model the true interventional quantity (the left side). Our use of "is equivalent to" (L199) referred to this idealized outcome. We will revise the text to explicitly state the theoretical conditions for this equality, clarifying it as the target of our optimization. Thank you for helping us improve the paper's precision.
>
> > Q3: L202-203.
>
> To be precise, the suppression of confounding effects is achieved by the cross-attention between prompts and image features, not by the prompts acting alone. The prompts simply provide the "directions" for this mechanism. We acknowledge the description in L202-203 was a simplification and will revise it for clarity. Thank you.
>
> > Q4: L220.
>
> We apologize for the ambiguity. You are correct about L220: we meant the visual prompts after the Cross-Attention Prompts, not the prompt parameter `p` itself. We will revise the text to make this distinction explicit.
>
> > Q5: The 'w/o DINOv2' ablation.
>
> In the 'w/o DINOv2' case, the backbone was a ResNet-50 (R50) pre-trained on ImageNet-1K, integrated within the DINO detector.
>
> [1] A fourier-based framework for domain generalization. CVPR 2021.
>
> [2]  Reducing domain gap by reducing style bias. CVPR 2021.

---

> > ### Comment · Reviewer_vcyR · 2025-08-04
> >
> > I appreciate the authors’ feedback.
> > I have read the feedback as well as the discussion with other reviewers.
> > Overall, my concerns are mostly addressed and I am remain positive about the paper.
> >
> > I have an additional question regarding equation 11.
> > I understood the intention behind the design of equation 10 and 11, but I still do not understand why the intended purpose is achieved by applying inverse Fourier transformation right after Fourier transformation without doing anything in the frequency domain. For example, [A1-A3] perform some operations such as masking after Fourier transformation, and thus it is understandable that the intended objectives are realized.
> > For this, I would like the authors to confirm that equation 10 and 11 is applied sequentially, i.e., equation 11 is applied right after equation 10.
> >
> > [A1] A Fourier-based framework for domain generalization. CVPR 2021.
> > [A2] Spectral invariant learning for 384 dynamic graphs under distribution shifts, NeurIPS 2023.
> > [A3] FDA: Fourier domain adaptation for semantic segmentation, CVPR 2020.
> >
> > I apologize for overlooking the rule that no images could be added in this phase.
> > However, the results provided by the authors have addressed my concerns.

---

> > > ### Author Response · Authors · 2025-08-04
> > >
> > > Dear Reviewer vcyR:
> > >
> > > We thank you for raising this valuable question, which gives us the opportunity to clarify a misunderstanding in the paper.
> > >
> > > > Q1: Are Equations 10 and 11 applied sequentially?
> > >
> > > **Yes — Equation (10) and Equation (11) are executed sequentially, but the output of Equation (11) is ultimately injected back in the form of a residual.**
> > >
> > > Equation (10) applies an MLP transformation to the features, whose output is then passed to the Fourier transform (Equation (11)) for an `FT → IFT` operation. It is important to emphasize that this Fourier transform exists **in the form of a residual**, as shown in our code.
> > > ```python
> > > def forward(self, x):
> > >     x = self.mlp(x)   # Eq. (10)
> > >     out = self.fourier_transform(x) # Eq. (11)
> > >     return out + x
> > > ```
> > >
> > > Therefore, the Fourier transform is **complementary and auxiliary** rather than the primary feature extraction branch. We apologize for any confusion caused by the notation and will revise it in the revised version to clearly indicate the residual nature of its structure.
> > >
> > > > Q2: Why is the intended purpose achieved by applying an inverse Fourier transform directly after a Fourier transform without any frequency-domain operations?
> > >
> > > **To directly answer your question**: there are two main reasons why we do not perform explicit transformations in the frequency domain. 1\) **Ablation study–driven**: Experimental results (Table 4 in our paper) show that adding explicit frequency-domain operations (e.g., masking) not only fails to provide any benefit but can even undermine its function as an auxiliary residual branch. 2\) **Design rationale**: This transform is intended to provide implicit spectral regularization and information compensation, rather than relying on explicit frequency-domain operations such as masking to improve performance on the source domain.
> > >
> > > 1\) In the ablation study (see Table 4 in our paper), we also experimented with explicitly adding a mask in the frequency domain. However, the results did not improve performance as expected, as shown in the table below. We believe this is because **the Causal Branch already possesses strong feature modeling capabilities**, while in the Auxiliary Branch, the Fourier transform serves only an auxiliary role rather than being the primary feature extraction branch. Due to its auxiliary role, explicit frequency-domain masking has only a limited impact on the final performance. Therefore, within the Dual-branch Adapter, the effect of frequency-domain masking is naturally unlikely to produce significant performance gains, since it is not the core factor driving improvement.
> > > |Function|DC|DF|DR|NR|NC|Avg.|
> > > |-|-|-|-|-|-|-|
> > > |w/o FFT|73.0|55.0|62.9|**48.5**|59.3|59.8|
> > > |Add Mask|**73.7**|56.2|64.1|48.2|60.1|60.5|
> > > |**Ours**|73.7|**56.5**|**64.6**|47.6|**61.2**|**60.7**|
> > >
> > > 2\) In contrast to [A1–A3], our design of the Fourier transform aims to introduce an **implicit spectral constraint**, enabling the model to learn stable features that are insensitive to frequency-domain perturbations. In this context, the `FT → IFT` operation serves a **regularization and information-compensation role during training**. However, introducing explicit frequency-domain operations such as masking would block the transmission of certain frequency components, **weakening the information-compensation role of the entire Auxiliary Branch**, and could even cause an obstruction of the overall information flow, thereby undermining its intended design as an auxiliary residual branch.
> > >
> > > We sincerely hope this explanation resolves your concern, and we are very grateful for your careful and thorough review and constructive feedback. Should you have any further questions or suggestions, please feel free to let us know. Once again, thank you for your valuable comments, which have been instrumental in helping us improve the quality of our work.
> > >
> > > With best regards, Authors.
> > >
> > > [A1] A Fourier-based Framework for Domain Generalization. CVPR 2021.
> > >
> > > [A2] Spectral Invariant Learning for Dynamic Graphs under Distribution Shifts. NeurIPS 2023.
> > >
> > > [A3] FDA: Fourier Domain Adaptation for Semantic Segmentation. CVPR 2020.

---

> > > > ### Comment · Reviewer_vcyR · 2025-08-04
> > > >
> > > > I sincerely appreciate the authors' prompt and thorough explanations despite the limited time. Their responses have been very helpful in improving my understanding.
> > > > In particular, the provided code has greatly made the discussion smoother.
> > > >
> > > > My concern is as follows: if we consider `self.fourier_transform(x)` to simply represent the Fourier transform and its inverse, then the result of applying it (i.e., `out`) would remain unchanged from `x` and thus be identical to `x`.
> > > > I may be misunderstanding something on this point, so I would greatly appreciate further clarification.

---

> > > > > ### Author Response · Authors · 2025-08-05
> > > > >
> > > > > Dear Reviewer vcyR,
> > > > >
> > > > > Thank you for the prompt and constructive feedback. We would like to first clarify that in our implementation, the operation is **not** an identity mapping—`out` is fundamentally different from `x` due to a **critical step—taking the real part**:
> > > > > ```python
> > > > > ifft_feats = torch.fft.ifft(fft_feats, dim=1).real  # take real part
> > > > > ```
> > > > >
> > > > > 1\) **Engineering necessity**: In deep learning, models operate in the real domain. The inverse FFT produces complex‑valued signals. Directly propagating complex numbers is not supported in PyTorch, causing gradient/backprop issues and large memory overhead. Applying `.real` is therefore **mandatory**, not optional, and discards the imaginary part (phase information) while **retaining the real part (magnitude information)**, thereby breaking the identity mapping at the signal level.
> > > > >
> > > > > 2\) **Causal relevance**: In the 2D Fourier spectrum, the **magnitude spectrum** encodes stable structural cues (shape, dominant textures), while the phase spectrum determines precise spatial alignment that is easily disturbed by illumination changes, weather variations, or noise. Discarding phase via `.real` preserves stable, causal-invariant information and filters out high-variance non-causal noise. FDA [1] aligns styles by swapping **low-frequency magnitude spectra** between source and target, significantly reducing domain gaps and confirming the key role of low-frequency magnitude in cross-domain structural invariance.
> > > > >
> > > > > Therefore, in our engineering implementation, **by retaining the magnitude while discarding the phase information**, the FFT output is already substantially altered in its signal characteristics. As a result, no additional complex operations in the frequency domain are required for it to serve as a valuable complement to the Causal Branch. We will clarify this distinction in the revised version. Thank you again for the insightful question and for helping us improve the clarity and quality of our work. We sincerely hope that our detailed clarifications address your concern.
> > > > >
> > > > > With best regards,
> > > > > Authors
> > > > >
> > > > > [1] FDA: Fourier Domain Adaptation for Semantic Segmentation. CVPR 2020.

---

> > > > > > ### Comment · Reviewer_vcyR · 2025-08-07
> > > > > >
> > > > > > Thank you again for the explanation.
> > > > > >
> > > > > > Assuming that `out = ifft_feats = torch.fft.ifft(fft_feats, dim=1).real`, it still appears to be an identity mapping to me, so I remain confused.
> > > > > > e.g.,
> > > > > > ```
> > > > > > x = torch.randn(10)
> > > > > > fft_feats = torch.fft.fft(x)
> > > > > > ifft_feats = torch.fft.ifft(fft_feats).real
> > > > > >
> > > > > > x: tensor([-0.7516, -1.2641,  0.3750,  0.5544,  2.1979,  2.8577,  0.6314,  0.8554,
> > > > > >          0.9650,  0.7984])
> > > > > >
> > > > > > fft_feats: tensor([ 7.2195+0.0000j, -6.2967+1.1389j,  0.6120+3.6212j, -2.5346-0.0520j,
> > > > > >          1.0437+1.8583j, -0.3839+0.0000j,  1.0437-1.8583j, -2.5346+0.0520j,
> > > > > >          0.6120-3.6212j, -6.2967-1.1389j])
> > > > > >
> > > > > > ifft_feats: tensor([-0.7516, -1.2641,  0.3750,  0.5544,  2.1979,  2.8577,  0.6314,  0.8554,
> > > > > >          0.9650,  0.7984])
> > > > > > ```
> > > > > >
> > > > > > > In the 2D Fourier spectrum, the magnitude spectrum encodes stable structural cues (shape, dominant textures), while the phase spectrum determines precise spatial alignment that is easily disturbed by illumination changes, weather variations, or noise.
> > > > > >
> > > > > > I agree on this point.
> > > > > >
> > > > > > >  Discarding phase via .real preserves stable,
> > > > > >
> > > > > > What I do not understand is this point.
> > > > > >
> > > > > > Can you clarify this point, please?

---

> > > > > > > ### Author Response · Authors · 2025-08-08
> > > > > > > **Thank you for the further discussion.**
> > > > > > >
> > > > > > > Thank you for your valuable comments. Your careful observations and experiments helped us identify an overlooked aspect of our structural design. We must first acknowledge that our initial understanding of `.real` was influenced by the following code snippet:
> > > > > > > ```
> > > > > > > import torch
> > > > > > > torch.set_printoptions(precision=10, sci_mode=False)
> > > > > > > x = torch.randn(10)
> > > > > > > fft_feats = torch.fft.fft(x)
> > > > > > > ifft_feats = torch.fft.ifft(fft_feats).real
> > > > > > > print(x == ifft_feats) # [False, False,  True, False, False, False, False,  True, False, False]
> > > > > > > x [-0.4920912981, -0.2312953621, -1.9472221136, -0.8723965287,
> > > > > > >          0.3732707798, -0.8995720744, -0.2371343225, -0.6218926907,
> > > > > > >         -0.1697061062, -0.6967265606]
> > > > > > > ifft_feats [-0.4920912683, -0.2312953025, -1.9472221136, -0.8723964691,
> > > > > > >          0.3732709289, -0.8995721936, -0.2371342927, -0.6218926907,
> > > > > > >         -0.1697060615, -0.6967265010]
> > > > > > > ```
> > > > > > >
> > > > > > > In fact, the observed numerical differences are negligible. Considering the domain generalization context, we initially hypothesized that the domain gap or the use of randomly learnable prompts might amplify this small difference, thus making `.real` have a substantial effect.
> > > > > > >
> > > > > > > In our earlier response—based on that prior assumption—we believed `.real` played a key role in this structure. However, after conducting more fine-grained experiments as you suggested, we found that `.real` itself does not substantively affect the amplitude or structural properties of the signal; it mainly serves as an engineering step to ensure a real-valued output. Your conclusion is correct: without other processing, `ifft(fft(x))` is essentially an identity mapping. The actual performance improvement is determined by the subsequent frequency-domain normalization (`norm`) operation:
> > > > > > > ```
> > > > > > > fft_feats = fft_feats / (torch.norm(fft_feats, dim=1, keepdim=True) + 1e-10)
> > > > > > > ifft_feats = torch.fft.ifft(fft_feats, dim=1).real
> > > > > > > ```
> > > > > > >
> > > > > > > A straightforward example: if $ x_2 = 2x_1$, then their spectra satisfy $ F_2 = 2F_1$. After normalization, we have
> > > > > > >
> > > > > > > $$
> > > > > > > \tilde{F}_1 = \frac{F_1}{\|F_1\|_2}, \quad
> > > > > > > \tilde{F}_2 = \frac{F_2}{\|F_2\|_2} = \frac{2F_1}{2\|F_1\|_2} = \tilde{F}_1
> > > > > > > $$
> > > > > > > as shown in the code below:
> > > > > > > ```
> > > > > > > import torch
> > > > > > > x1 = torch.randn(8)
> > > > > > > x2 = 2 * x1
> > > > > > > F1 = torch.fft.fft(x1)
> > > > > > > F2 = torch.fft.fft(x2)
> > > > > > > F1n = F1 / torch.norm(F1)
> > > > > > > F2n = F2 / torch.norm(F2)
> > > > > > > print(torch.allclose(F1n, F2n))  # True
> > > > > > > print(torch.allclose(torch.fft.ifft(F1n), torch.fft.ifft(F2n)))  # True
> > > > > > > ```
> > > > > > >
> > > > > > > While iFFT alone acts as an identity mapping, in our implementation it is combined with global L2 normalization in the frequency domain, producing time-domain signals with constant overall energy (up to an FFT-dependent constant). This makes the branch **insensitive to global amplitude changes**, outputting the direction rather than the length of features, thereby achieving amplitude standardization.
> > > > > > >
> > > > > > > This standardization retains Fourier amplitude information while reducing phase-induced high-variance noise: in our setting, the amplitude spectrum largely encodes domain-specific style and global intensity statistics, whereas the phase spectrum—though encoding precise spatial alignment—is more vulnerable to illumination, weather, and noise. Frequency-domain normalization explicitly separates amplitude and phase, allowing the model to focus on stable structural cues while suppressing task-irrelevant cross-domain intensity variations.
> > > > > > >
> > > > > > > By Parseval’s theorem [1], global L2 normalization in the frequency and time domains are mathematically equivalent up to a constant determined by the FFT convention. We choose the frequency-domain formulation for explicit amplitude/phase separation and its flexibility for extensions such as band-wise normalization or spectral gating.
> > > > > > >
> > > > > > > In the SDGOD setting (day-clear → various weather), amplitude differences are mainly caused by illumination, haze, and contrast—factors that change global energy but are unrelated to object geometry or semantics. Frequency-domain amplitude normalization suppresses these biases, directing the model toward cross-domain stable relative spectral shapes and structural features.
> > > > > > >
> > > > > > > We sincerely acknowledge that `.real` is merely an engineering step for real-valued output and has no substantive effect on the amplitude or structure of the signal. The true contributor to the observed performance gain is the subsequent frequency-domain normalization, which adjusts the amplitude spectrum in the complex modulus sense, ensuring global amplitude invariance across domains. This insight—prompted by your feedback—enabled us to more accurately identify the actual mechanism behind this branch’s effectiveness. We will include in the revised manuscript an ablation study replacing frequency-domain L2 normalization with time-domain L2 normalization, along with reproducible code to verify amplitude invariance.
> > > > > > >
> > > > > > > [1] A. V. Oppenheim and R. W. Schafer, *Discrete-Time Signal Processing*, 3rd ed., Prentice Hall, 2009.

---

> > > > > > > > ### Comment · Reviewer_vcyR · 2025-08-08
> > > > > > > >
> > > > > > > > Thank you for the clarification.
> > > > > > > >
> > > > > > > > Now my questions have been resolved through the discussion. I believe clarifying what fundamentally contributes to performance improvement will be a contribution to the community. Please ensure that these discussions are indeed reflected in the manuscript.

---

### Note · Authors · 2025-08-13

Dear Reviewers and Area Chairs,

We would like to express our sincere gratitude to the reviewers for their constructive feedback and insightful comments.

Our work presents Cauvis, a novel approach for domain generalization in object detection that disentangles causal from spurious correlations through cross-attention prompts and a dual-branch adapter, enabling the network to focus on robust features while ignoring spurious features.

# Discussion of Rebuttal

We addressed key concerns from the reviewers by clarifying several aspects of Cauvis. For Reviewer `vcyR`, we experimentally validated the auxiliary branch’s theoretical design and practical value, and, for the first time via singular value decomposition, quantitatively showed that Cauvis effectively separates causal from spurious features. Subsequent discussions also clarified the design mechanism of the Fourier transform. For Reviewer `L5GS`, we demonstrated that prompt and adapter tuning effectively separate causal and spurious correlations, improving domain generalization by up to 2.9% with minimal computational overhead, and confirmed compatibility with diverse backbones. For Reviewer `soZz`, we showed that Cauvis significantly boosts performance with negligible cost, achieving strong cross-dataset generalization, especially from synthetic to real scenarios, and maintaining robustness under extreme weather conditions, highlighting its portability. For Reviewer  `pbD8`, we clarified the distinction between SDGOD and DA, showing that without target-domain data, Cauvis achieves superior synthetic-to-real performance, and is the first to apply DINOv2 to SDGOD, proving its suitability and enhancing Cauvis’s strong generalization.

# Integration of DINOv2 for SDGOD

Regarding the contribution of introducing DINOv2 into SDGOD, we would like to clarify that this application is indeed a contribution. Although large-scale pre-trained models such as DINOv2 are common in other areas, this is the first time it has been applied to the SDGOD task. We have demonstrated that DINOv2 not only significantly improves the accuracy of the baseline model but also highlights the unique design of Cauvis and the advantages of its integration with DINOv2. As the trend of employing vision-language models in existing work continues to grow, we believe our study offers a new perspective for selecting foundation models and proves that DINOv2 is highly suitable for this task, providing valuable insights to the community.

---

### Decision · Program_Chairs · 2025-09-17

**Decision:**

Accept (poster)

**Comment:**

This paper introduces Cauvis, a method for Single-Source Domain Generalized Object Detection (SDGOD) that tackles the critical challenge of spurious correlations. The core of its contribution lies in using a cross-attention prompt module, inspired by causal inference, and a dual-branch adapter to disentangle domain-invariant causal features from spurious ones. The strengths of this work, as highlighted by the reviewers, include its clear motivation, sound theoretical rationale, and extensive experiments demonstrating state-of-the-art performance with significant gains over existing methods. Initial weaknesses and questions were raised by the reviewers, including concerns about the justification for the dual-branch adapter's design (vcyR), the fairness of comparisons given the strong DINOv2 backbone (pbD8), computational overhead (L5GS), and the need for more extensive validation on diverse benchmarks and source domains (soZz). The authors provided a comprehensive and convincing rebuttal, effectively addressing all major points. They clarified the training procedure, provided new quantitative experiments to validate the disentanglement of features, and engaged in an in-depth discussion that ultimately clarified the role and mechanism of the Fourier transform component in the adapter, satisfying reviewer vcyR. They also conducted new experiments on additional corruption benchmarks and with harsh-weather source domains, demonstrating the method's robustness and generality, which addressed the concerns of reviewer soZz. Furthermore, they successfully justified the performance gains over a very strong baseline and contextualized their contribution regarding the use of large foundation models, addressing the concerns of reviewers L5GS and pbD8. Based on the reviews and discussion, AC recommends accept.